# Constructing regulable supports via non-stoichiometric engineering to stabilize ruthenium nanoparticles for enhanced pH-universal water splitting

Sheng Zhao [1], Sung-Fu Hung [2], Liming Deng[1], Wen-Jing Zeng [2], Tian Xiao[1], Shaoxiong Li [1], Chun-Han Kuo[3], Han-Yi Chen [3], Feng Hu[1] & Shengjie Peng [1] ✉

Establishing appropriate metal-support interactions is imperative for acquiring efficient and corrosion-resistant catalysts for water splitting. Herein, the interaction mechanism between Ru nanoparticles and a series of titanium oxides, including TiO, $Ti_4O_7$ and $TiO_2$, designed via facile non-stoichiometric engineering is systematically studied. $Ti_4O_7$ with the unique band structure, high conductivity and chemical stability, endows with ingenious metal-support interaction through interfacial Ti–O−Ru units, which stabilizes Ru species during OER and triggers hydrogen spillover to accelerate HER kinetics. As expected, $Ru/Ti_4O_7$ displays ultralow overpotentials of 8 mV and 150 mV for HER and OER with a long operation of 500 h at 10 mA cm$^{-2}$ in acidic media, which is expanded in pH-universal environments. Benefitting from the excellent bifunctional performance, the proton exchange membrane and anion exchange membrane electrolyzer assembled with $Ru/Ti_4O_7$ achieves superior performance and robust operation. The work paves the way for efficient energy conversion devices.

Water splitting has received extensive concern as the most promising pathway to obtain green hydrogen[1,2]. Membrane electrode assembly (MEA) water electrolysis allows for clean and efficient green hydrogen production[3,4]. Anion exchange membrane (AEM) and proton exchange membrane (PEM) electrolyzers are promising devices for water electrolysis, which rely on platinum (Pt)-based catalysts for hydrogen evolution reaction (HER)[5–7]. Furthermore, PEM electrolyzers also require corrosion-resistant iridium (Ir)-based materials due to the harsh acidic conditions for oxygen evolution reaction (OER)[8,9]. The large-scale consumption of precious metals has resulted in high costs for PEM, hindering the commercialization of PEM water electrolysis[10,11]. Therefore, it is necessary to develop low-cost catalysts with less or non-Pt/Ir based catalysts. Ru is an effective alternative to Pt/Ir due to about

1/5 of the cost of Pt[12]. More importantly, Ru NPs exhibit unique HER and OER bifunctionality advantages in water splitting to reduce the catalyst cost[12–14]. However, the stronger affinity of Ru for the *H intermediate in HER significantly affects reaction kinetics[15]. In the OER process, Ru easily forms soluble $Ru^{n+}$ species ($n > 4$) under the oxidation potentials, thus exhibiting an extremely short lifetime[16,17]. Enhancing the activity and stability of Ru NPs in both HER and OER is a challenge. Among many strategies of catalyst design, loaded catalysts have garnered widespread attention due to the simple synthesis process, low usage of precious metals, ample exposure of active sites, and high atomic utilization efficiency[18,19]. More importantly, sufficient contact between the support and active phase can construct suitable metal-support interactions and stable interface units to regulate the coordination

[1]College of Materials Science and Technology, Nanjing University of Aeronautics and Astronautics, Nanjing 210016, China. [2]Department of Applied Chemistry, National Yang Ming Chiao Tung University, Hsinchu 300, Taiwan. [3]Department of Materials Science and Engineering, National Tsing Hua University, Hsinchu 30013, Taiwan. ✉e-mail: pengshengjie@nuaa.edu.cn

configuration and electronic structure of interface sites, which optimizes the catalytic performance and stability of the active centers[20–22].

In general, ideal electrocatalyst support needs to maintain stability with high electroconductibility to minimize excessive energy consumption and catalyst dissolution under the operating voltages[23–27]. Among the common stabilized supports, such as $TiO_2$[20,28], $MnO_2$[29], $WO_3$[15,30], $MoO_3$[31], etc., $TiO_2$ has attracted much attention due to its excellent corrosion resistance and adjustable valence[20]. However, $TiO_2$ usually displays a broad energy band gap as a typical n-type semiconductor material, which gives it poor conductivity and results in limited application in electrocatalysis[32]. Introducing appropriate oxygen vacancies into $TiO_2$ through defect engineering can create defect energy levels to improve the conductivity[28,33]. Nevertheless, random oxygen vacancy distribution makes it difficult to adjust defect states accurately and causes potential structural imbalances in semiconductors. Therefore, constructing non-stoichiometrically stable oxide supports with periodically distributed defects can introduce lattice defects with stable structures. Non-stoichiometric titanium oxides ($Ti_nO_{2n-1}$), such as $TiO$[34], $Ti_3O_5$[35], and $Ti_4O_7$[34], possess higher electrical conductivity for several orders of magnitude than $TiO_2$[28,34,36]. In particular, $Ti_4O_7$, as a typical non-stoichiometric titanium oxide, has a similar conductivity to that of graphene, which is attributed to the abundant density of state distribution at the Fermi level[37]. In addition, $Ti_4O_7$ exhibits a more stable structure compared to lower-valence titanium oxides such as $TiO$. More importantly, the precise non-stoichiometric design of the support can obtain the suitable crystal and electronic structure to achieve adaptability with the active species, which is beneficial to improve the intrinsic activity and stability of the active sites.

Herein, the interaction mechanism between Ru nanoparticles and a series of titanium oxides, including $TiO$, $Ti_4O_7$ and $TiO_2$, was systematically explored. The appropriate metal-support interaction between Ru and $Ti_4O_7$ is achieved, which moderately induces electron enrichment of the Ru sites to inhibit the lattice oxygen mechanism (LOM) of OER and facilitate deprotonation through stable Ti–O–Ru units, which balances the activity and stability of OER. Furthermore, the low interface resistance between $Ti_4O_7$ and Ru initiates the hydrogen spillover mechanism for HER to accelerate reaction kinetics. The exceptional bifunctional activity is confirmed in pH-universal environments. Specifically, in acidic conditions, Ru/$Ti_4O_7$ demonstrates ultra-low overpotentials of 8 mV and 150 mV at 10 mA cm$^{-2}$ for HER and OER, respectively, and maintains durable operation for 500 h. PEM devices assembled with Ru/$Ti_4O_7$ show lower cell voltages and longer stability than those of commercial $RuO_2 \parallel$ Pt/C. This work provides new insight into the development of rationally supported catalysts and paves the way for the design of energy conversion devices.

## Results

### Principles of carrier design

As shown in Fig. 1, non-stoichiometric engineering was employed to finely customize titanium oxide supports ($TiO$, $Ti_4O_7$, and $TiO_2$) with oxygen vacancies of periodic arrangements. $Ti_4O_7$, with regularly arranged defects, emerges as a potential competitor with superior structural stability over multi-defect $TiO$ and significantly higher conductivity than $TiO_2$. More importantly, $Ti_4O_7$ exhibits a suitable metal-support interaction, which lays the foundation for the dual-function activity of Ru/$Ti_4O_7$ in both HER and OER. On the one hand, the higher work function compared to Ru NPs promotes the electron richness of Ru, which can alleviate the dissolution of Ru in OER. On the other hand, compared to $TiO$ and $TiO_2$, the minimum work function difference between $Ti_4O_7$ and Ru NPs ($\Delta\Phi = 0.30$ eV) can reduce the interface Schottky barrier (Supplementary Fig. 1), which promotes electron transport in the composite catalyst and trigger hydrogen spillover during HER.

## Morphology and crystal structure

Ru nanoparticles (NPs) were deposited on titanium oxide supports with different stoichiometric ratios ($TiO_2$, $Ti_4O_7$, and $TiO$) via a wet chemical method (Fig. 2a). $Ti_4O_7$ has potential electrocatalytic advantages in support of Ru NPs due to the chemical stability superior to $TiO$ and electrical conductivity far superior to $TiO_2$. Scanning electron microscopy (SEM) and transmission electron microscopy (TEM) images display the uniform coating of the titanium oxide supports by Ru NPs with a size of ~10 nm (Figs. 2b and 1c). The suitable oxygen vacancies in $Ti_4O_7$ promote the $Ru^{3+}$ adsorption on $Ti_4O_7$, which endows the Ru NPs with a more uniform distribution on $Ti_4O_7$ compared to $TiO_2$ with low vacancies (Supplementary Figs. 2–6)[38,39]. In aberration-corrected high-resolution (AC-HRTEM) of Ru/$Ti_4O_7$, tight binding and appropriate matching at the nanoscale between the different phases can be observed, accompanied by a smooth transition at the interface (Fig. 2d). The simulated High Angle toroidal dark field image-scanning transmission electron microscope (HAADF-STEM) images obtained through the crystal structure of $Ti_4O_7$ and Ru show remarkable agreement with the experimental results (Figs. 2e and 1f). Based on these atomic images, the lattice spacing of 0.377 nm can be attributed to the (1 0 2) plane of $Ti_4O_7$ (JCPDS 50-0787), while the lattice spacing of 0.234 nm corresponds to the (1 0 0) plane of Ru (JCPDS 89-4903, Figs. 2g, 1h, and Supplementary Fig. 7). In addition, Ru, Ti, and O elements are uniformly distributed in the energy-dispersive spectroscopy (EDS)-mapping of Ru/$Ti_4O_7$ (Fig. 2i), and the characteristic diffraction peaks assigned to Ru and $Ti_4O_7$ coexist in the X-ray diffraction (XRD) pattern (Fig. 2j). The above results indicate that $Ti_4O_7$ can achieve a suitable interfacial lattice contact with Ru NPs, which provides the possibility for adaptive metal-support interaction and rapid interface electron migration[21,40].

## Electronic and coordination structure analysis

The oxygen vacancy concentration in Ru/$TiO_2$, Ru/$Ti_4O_7$, and Ru/$TiO$ shows an increasing trend due to the different stoichiometric ratios of Ti and O (Fig. 3a). The periodically arranged oxygen vacancy not only improves the conductivity of the support but also provides the possibility to form the bridging interfaces between the $Ti_4O_7$ and the Ru NPs. In the X-ray photoelectron spectroscopy (XPS) fine spectra of Ru/$TiO_2$, Ru/$Ti_4O_7$, and Ru/$TiO$, the Ti 2$p$ and Ru 3$p$ regions display close overlap (Fig. 3b). The binding energy of Ti 2$p$ in different Ru-loaded titanium oxide materials shows various degrees of shift to the high binding energy (Fig. 3b, and Supplementary Figs. 8–10), which indicates that titanium oxide can serve as an electron donor to promote electron enrichment of Ru (Fig. 3c). As shown in Supplementary Fig. 8c, Ru NPs exhibit only the characteristic peaks of metallic Ru. In the Ru 3d region of Ru/$Ti_4O_7$, the signals attributed to Ru–O and Ru–Ru can be observed, corresponding to the interface Ti–O–Ru units[41,42]. Similar local electronic changes can be further observed in the fine structure of X-ray absorption (XAFS). In the X-ray absorption near-edge structure (XANES), the Ru $K$-edge also exhibits different white line peak intensity after contacting with various titanium oxide supports (Fig. 3d). Importantly, Ru in Ru/$Ti_4O_7$ has the closest valence state to Ru foil compared with that in Ru/$TiO_2$ and Ru/$TiO$, which is attributed to the low work function difference of $Ti_4O_7$ and Ru obtained by the ultraviolet photoemission spectroscopy (UPS) measurements (Supplementary Fig. 1). Meanwhile, Ti pre-$K$-edge of Ru/$Ti_4O_7$ shows a shift to higher energy compared with that of $Ti_4O_7$, suggesting that the increased valence state of Ti in Ru/$Ti_4O_7$ (Supplementary Fig. 11). The above results indicate that Ru serves as an electron acceptor to form a Schottky contact with the titanium oxide support, and $Ti_4O_7$ possesses the best adaptability of the electronic structure with Ru NPs to reduce the interface Schottky barrier. Extended X-ray absorption fine structure (EXAFS) spectra of Ru corresponding k$^2$-weighted EXAFS Fourier transform spectra display a weak Ru–O signal located at 1.92 Å after contacting with $TiO_2$ or $Ti_4O_7$

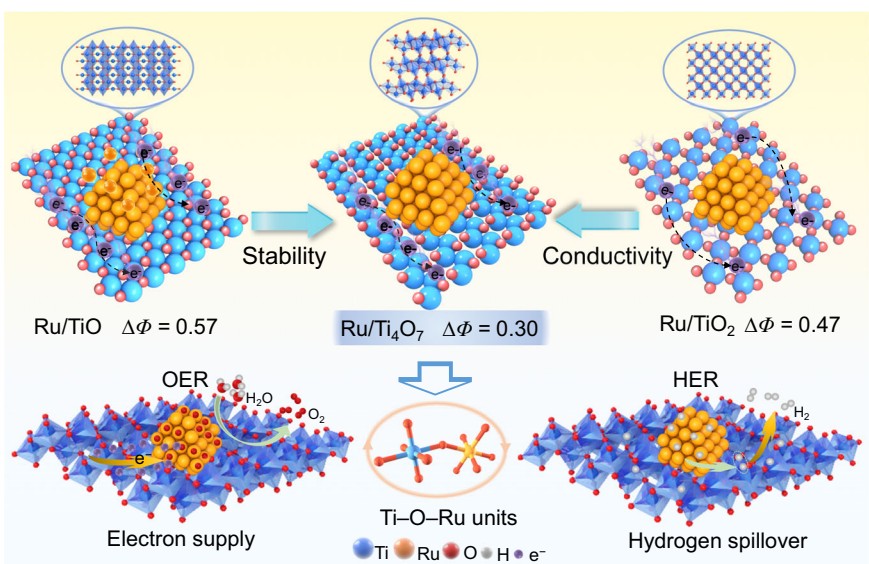

**Fig. 1 | Bifunctional activation based on different supports.** Schematic diagram of the interaction mechanism between different titanium oxide supports and Ru NPs to activate OER and HER.

**Fig. 2 | Synthesis and structural characterization. a** SEM of Ru/Ti$_4$O$_7$. **b** TEM and **c** AC-HRTEM images of Ru/Ti$_4$O$_7$. **d**, **e** The schematic atom structure and the corresponding simulated HAADF-STEM images of Ti$_4$O$_7$ and Ru in Ru/Ti$_4$O$_7$, respectively. **f** Crystal plane spacing of Ti$_4$O$_7$ and Ru. **g** Schematic diagram of interface contact between Ti$_4$O$_7$ and Ru. **h** EDS mapping of Ru, Ti, and O in Ru/Ti$_4$O$_7$, respectively. **i** XRD pattern of Ru/Ti$_4$O$_7$.

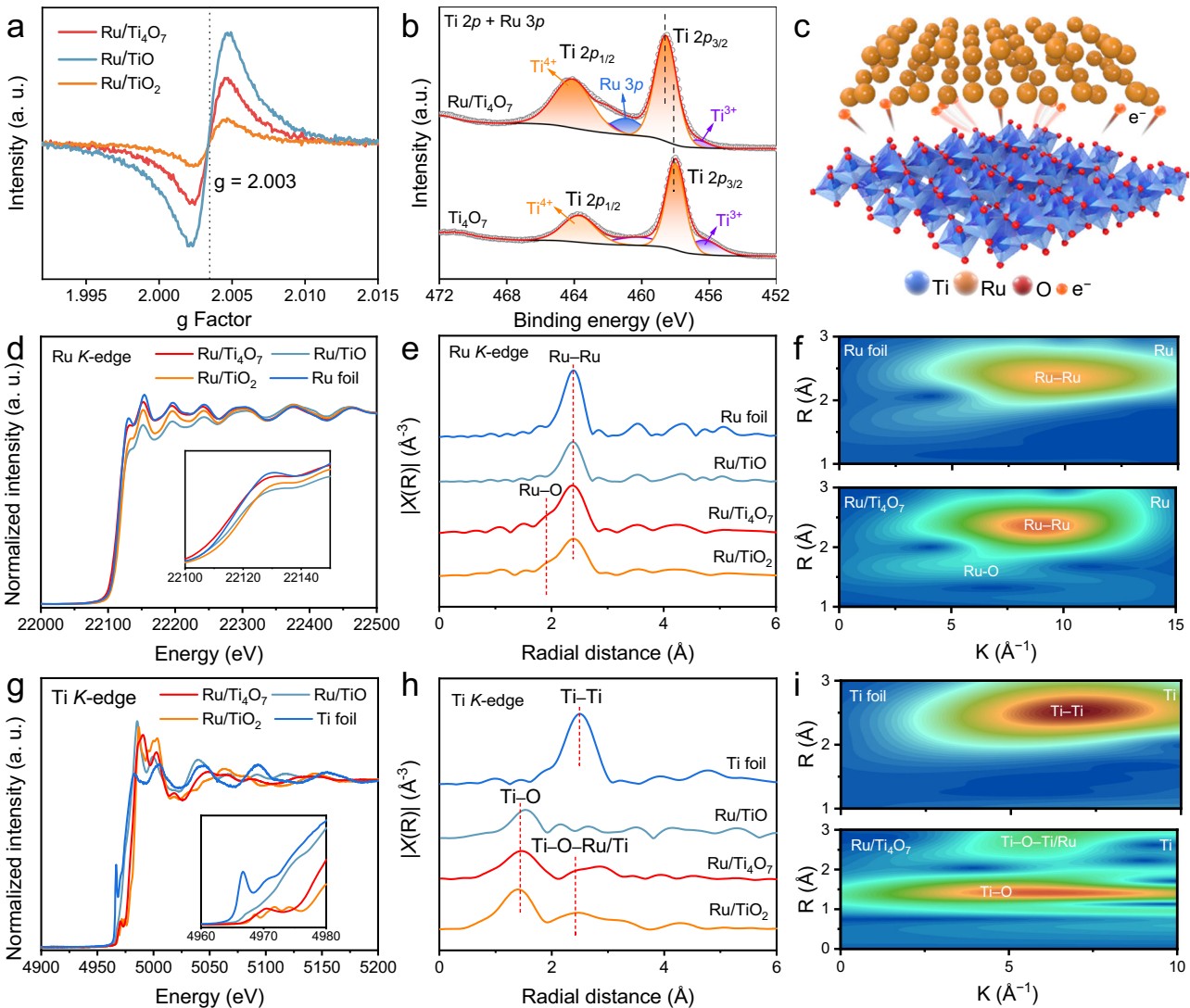

**Fig. 3 | Electron and coordination structure characterization. a** EPR spectra of Ru/TiO$_2$, Ru/Ti$_4$O$_7$, and Ru/TiO. **b** XPS fine spectra in Ti 2$p$ and Ru 3$p$ region of Ti$_4$O$_7$, and Ru/Ti$_4$O$_7$. **c** Schematic illustration of the electronic interaction between Ti$_4$O$_7$ and Ru. **d** Normalized Ti $K$-edge XANES of Ti foil, Ru/TiO$_2$, Ru/Ti$_4$O$_7$, and Ru/ TiO. **e** The corresponding k$^2$-weighted Fourier transforms and (**f**) Wavelet transform of k$^2$-weighted EXAFS signals. **g** Normalized Ru $K$-edge XANES of Ru foil, Ru/TiO$_2$, Ru/Ti$_4$O$_7$, and Ru/TiO. **h** The corresponding k$^2$-weighted Fourier transforms and **i** Wavelet transform of k$^2$-weighted EXAFS signals.

(Fig. 3e). This phenomenon should originate from the interface Ti–O–Ru units between Ru and titanium oxide, which is observed in the wavelet transform more clearly (Fig. 3f). The stable Ti–O–Ru units effectively stabilize the active species and serve as an electron channel to further reduce the interface electron transfer resistance[43]. As shown in Fig. 3g, the shift of Ti pre-$K$-edge reflects the changed oxide valence states of Ti in different stoichiometric titanium oxide supports. Furthermore, As illustrated in the k$^2$-weighted EXAFS Fourier transforms of Ti $K$-edge of Ru/TiO$_2$ and Ru/Ti$_4$O$_7$ (Fig. 3h) and the corresponding wavelet transform (Fig. 3i), the structure belonging to Ti–O–Ru/Ti further verifies the formation of the interface Ti–O–Ru units.

**Electrocatalytic activity and stability evaluation**
The electrocatalytic activities of the Ru-loaded titanium oxides were evaluated in a standard three-electrode system to verify the unique advantages of the Ti$_4$O$_7$ support. In the acidic media, Ru/Ti$_4$O$_7$ with optimized Ru loading and annealing temperature exhibits the lowest OER overpotential (150 mV at 10 mA cm$^{-2}$, Fig. 4a and Supplementary Fig. 12) and Tafel slope (41.26 mv dec$^{-1}$, Fig. 4b, c). For HER, Ru/Ti$_4$O$_7$ also displays an ultra-low overpotential of 8 mV at 10 mA cm$^{-2}$ in acidic

environments, which is superior to commercial Pt/C and other titanium oxide supported Ru NPs (Fig. 4d). Remarkably, as shown in Fig. 4e, the Tafel slope of Ru/Ti$_4$O$_7$ with minimum work function difference is only 21.24 mV dec$^{-1}$, which is significantly lower than the value of the conventional Volmer-Heyrovsky/Tafel mechanism (30 mV dec$^{-1}$)[44–46]. The low work function difference between metal and support is prone to trigger the hydrogen spillover mechanism during HER and provide a low interfacial Schottky barrier and fast interfacial electron migration[47]. Electrochemical impedance spectroscopy (EIS) further verifies the lowest electron-transfer resistance of Ru/Ti$_4$O$_7$ (Supplementary Fig. 13). Notably, the superior difunctional activity of Ru/Ti$_4$O$_7$ also exhibited scalability in the neutral and basic environments (Fig. 4c, f, and Supplementary Figs. 14–17). In addition, Ru/Ti$_4$O$_7$ exhibits the highest double-layer capacitance ($C_{dl}$) value (68.95 mF cm$^{-2}$) with the same Ru loading, suggesting that more exposed Ru active sites of the Ru/Ti$_4$O$_7$ compared to Ru/TiO$_2$ and Ru/TiO (Supplementary Figs. 18 and 19). The LSV polarization curves normalized by $C_{dl}$ and the mass of Ru indicate that the active Ru sites in Ru/Ti$_4$O$_7$ have the highest specific activities (Supplementary Fig. 20) and mass activity (Supplementary Fig. 21). The outstanding

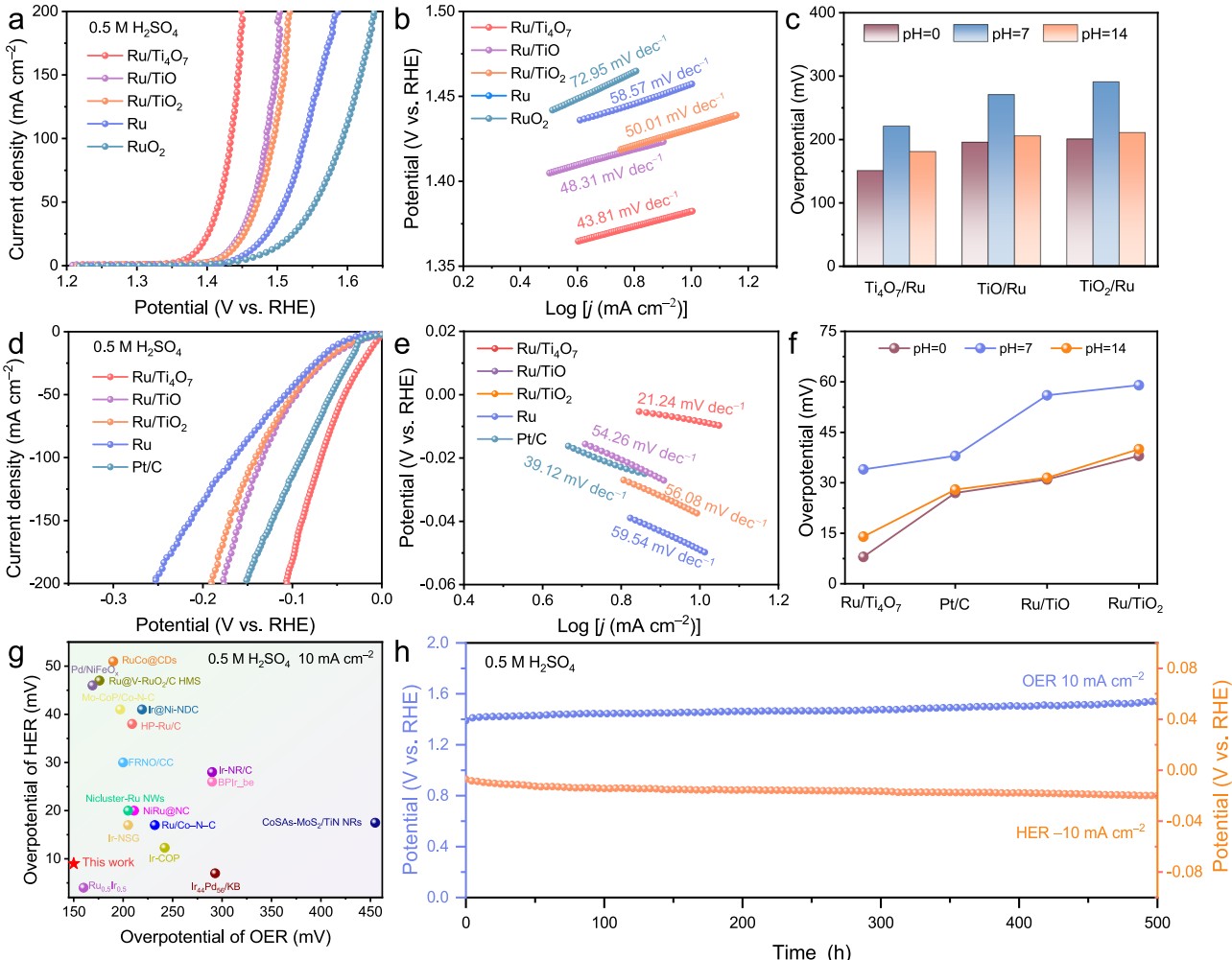

**Fig. 4 | Electrocatalytic activity evaluation. a** Linear sweep voltammetry (LSV) polarization curves of OER and **b** Tafel plots of Ru, commercial $RuO_2$, $Ru/TiO_2$, Ru/$Ti_4O_7$, and Ru/TiO (loading amount: 1 mg cm$^{-2}$) in 0.5 M $H_2SO_4$ (pH = 0.3). **c** The corresponding OER overpotential at 10 mA cm$^{-2}$ in different pH environments. **d** LSV polarization curves of HER and (**e**) Tafel plots of Ru, commercial Pt/C, Ru/ $TiO_2$, Ru/$Ti_4O_7$, and Ru/TiO (loading amount: 1 mg cm$^{-2}$) in 0.5 M $H_2SO_4$ (pH = 0.3). **f** The corresponding HER overpotential at −10 mA cm$^{-2}$ in different pH environments. **g** Comparison of overpotential with that of other bifunctional catalysts for water splitting in 0.5 M $H_2SO_4$ recently reported. **h** Chronopotentiometry curves of Ru/$Ti_4O_7$ at 10 mA cm$^{-2}$, and −10 mA cm$^{-2}$ in 0.5 M $H_2SO_4$.

bifunctional performance of Ru/$Ti_4O_7$ exceeds that of most of the excellent bifunctional catalysts in the recent reports (Supplementary Tables 2, 3), especially in acidic environments (Fig. 4g).

The stability of Ru/$Ti_4O_7$ was validated via chronopotentiometry. In severe acidic environments, Ru/$Ti_4O_7$ demonstrates surprising stability of OER and HER for 500 h with a negligible activity decay (Fig. 4h) and slight structural collapse (Supplementary Fig. 22), which exceeds the operating life of Ru/TiO (Supplementary Fig. 23). Ru/$Ti_4O_7$ also requires lower potentials to drive OER compared to Ru/$TiO_2$, contributing to the lower energy consumption of the water electrolysis devices over a long period (Supplementary Fig. 23). More importantly, Ru/$Ti_4O_7$ can maintain stable operation for 300 h at a high current density of 200 mA cm$^{-2}$ with only slight activity decay after 3000 cyclic voltammetry (CV) cycles (Supplementary Figs. 24 and 25). The lower current response in the CV corresponding to the oxidation-reduction of Ru in Ru/$Ti_4O_7$ compared to Ru and Ru/TiO implies the inhibited oxidation of Ru by $Ti_4O_7$ (Supplementary Fig. 26). Supplementary Fig. 27 manifests that the Ru and Ti loss of Ru/$Ti_4O_7$ gradually slows down during long-term OER. Ti gradually dissolves in the first 50 h of cycling, which is attributed to the reaction between the unstable surface of the electrode. After 50 h, the dissolution rate of Ti slows down due to the stabilization of the surface of the electrode. Furthermore,

the characteristic diffraction peaks belonging to Ru and $Ti_4O_7$ can still be observed in the XRD pattern of Ru/$Ti_4O_7$ after OER without new characteristic peaks compared to the apparent dissolution of Ru/TiO (Supplementary Fig. 28). As shown in the electron paramagnetic resonance (EPR) spectra (Supplementary Fig. 29) and XPS spectra of O 1$s$ before and after stability tests (Supplementary Fig. 30a), a slight decrease in oxygen vacancies in the $Ti_4O_7$ support is observed, indicating the structural stability of the $Ti_4O_7$ support under high current densities. In addition, the almost unchanged XPS signals of Ti and Ru with slight shift towards high binding energy after OER further provide evidence for the good structural maintenance of Ru/$Ti_4O_7$ during the reaction (Supplementary Fig. 30b). The excellent stability of Ru/$Ti_4O_7$ is attributed to inhibited oxidation process, which should be derived from the electron enrichment of Ru through the stabilized Ti−O−Ru units.

## Origin of enhanced difunctional activity of Ru/$Ti_4O_7$

A series of in-situ tests were performed in acid environments to trace the origin of the activity and stability of Ru/$Ti_4O_7$. In-situ XAFS explored the structural evolution of Ru/$Ti_4O_7$ during OER. Ru/$Ti_4O_7$ underwent 20 CV cycles to obtain the stable surface before recording the in-situ XAFS spectra. In the most severe acidic environments, the

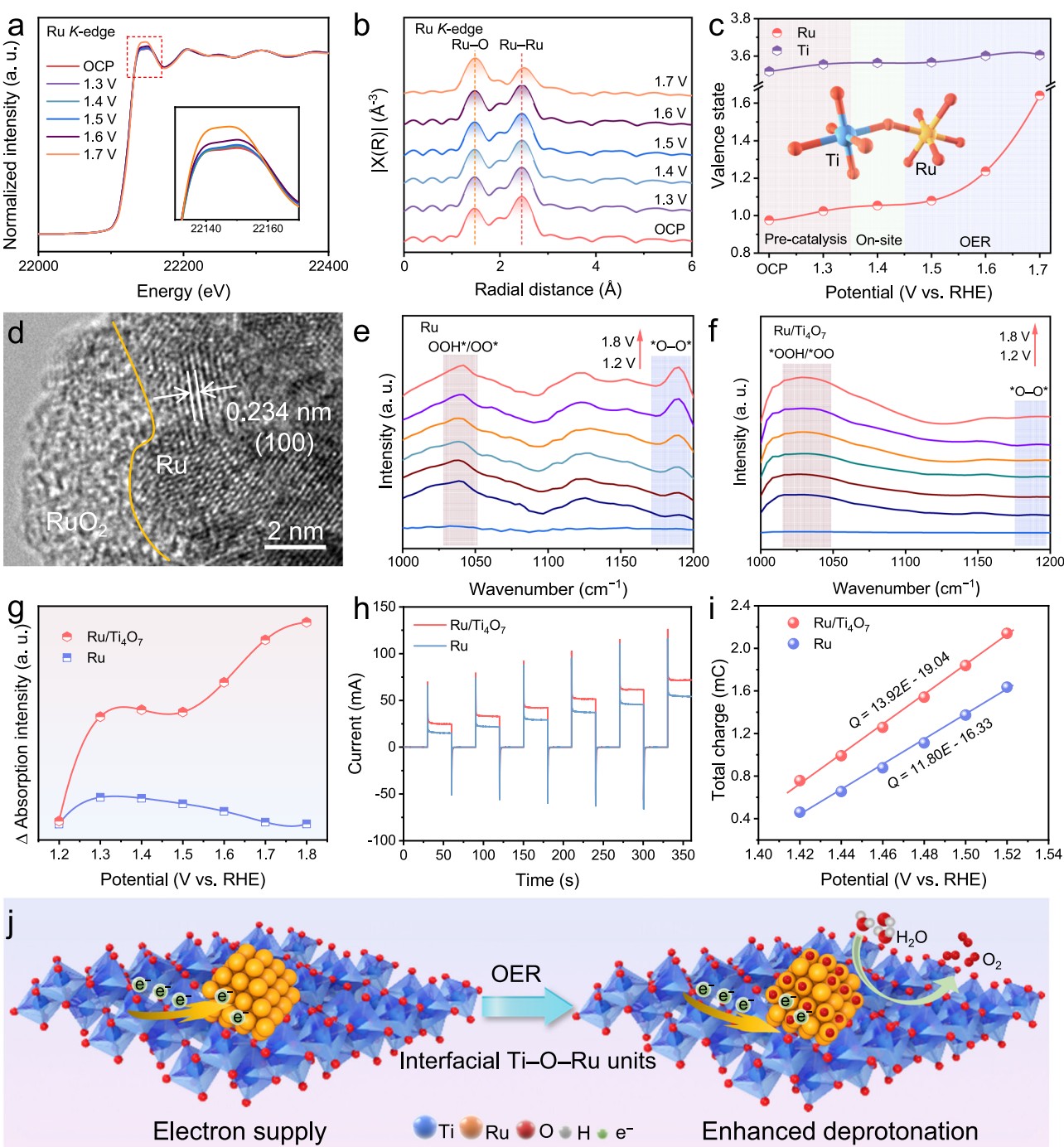

**Fig. 5 | In-situ characterization and reaction mechanism. a** Normalized in-situ Ru $K$-edge XANES of Ru/Ti$_4$O$_7$. **b** The corresponding k$^2$-weighted Fourier transforms. **c** The valence states of Ru and Ti in Ru/Ti$_4$O$_7$ obtained via Ru $K$-edge under different potentials. **d** HRTEM of Ru/Ti$_4$O$_7$ after OER. In-situ ATR-SEIRAS spectra of (**e**) Ru and **f** Ru/Ti$_4$O$_7$ for OER in 0.5 M H$_2$SO$_4$ under different potentials vs. RHE.

**g** Intensity difference of the infrared signals at 1038 and 1189 cm$^{-1}$. **h** Current responses to pulse voltammetry for Ru and Ru/Ti$_4$O$_7$. **i** Relationship between charge storage and potential of Ru and Ru/Ti$_4$O$_7$. **j** Schematic illustration of the optimized OER process induced by the Ti$_4$O$_7$ support.

intensity of the white line of the Ru $K$-edge increases with the applied potentials of OER (Fig. 5a), indicating the oxidation of Ru NPs. Furthermore, in the Fourier transform $R$-space (Fig. 5b), the gradually increased signal intensity of the Ru–O path verifies the formation of RuO$_x$. Notably, the peaks attributed to metallic Ru–Ru can always be observed even under the high potentials of OER, which corresponds to the Ru metallic phase shown in the XRD pattern of Ru/Ti$_4$O$_7$ after OER. The above results indicate that the RuO$_x$ species locally formed on the

Ru/Ti$_4$O$_7$ can act as a barrier layer to slow down the further oxidation and dissolution of Ru NPs. The in-situ XAFS spectra show that the Ti $K$-edge of Ti$_4$O$_7$ almost completely overlaps under the OER potentials without prominent oxidation characteristics, which benefits from the stable structure of Ti$_4$O$_7$ (Supplementary Fig. 31a). The $R$-space of Ti further verifies that the Ti–O bond in Ti$_4$O$_7$ does not change significantly during OER (Supplementary Fig. 31b). The stability of Ru/Ti$_4$O$_7$ can be more verified in the explicit valence states

of Ti, and Ru. The valence state corrected by the normalized white line intensity of the corresponding reference of Ti in Ru/Ti$_4$O$_7$ changes slightly between 3.52 and 3.61 with the application potentials growing (Supplementary Fig. 32). Meanwhile, the average valence state of Ru increases from 0.98 to 1.64. The significant change does not appear until the potential reaches 1.7 V vs. RHE (Fig. 5c). This phenomenon further verifies that the Ru NPs still maintain structural stability under high potentials of OER. The incomplete oxidation of Ru NPs is beneficial to maintaining the outstanding conductivity of the composite material during the electrocatalytic process. In the HRTEM image of Ru/Ti$_4$O$_7$ after OER, the tight interfacial contact between Ru NPs and Ti$_4$O$_7$ can still be maintained (Supplementary Fig. 33). In addition, a locally amorphous structure can be observed in the outer layer of Ru NPs in Ru/Ti$_4$O$_7$ after OER, which should correspond to the RuO$_x$ species (Fig. 5d).

Moreover, the reaction mechanism of OER was revealed by in-situ attenuated total reflection-surface enhanced infrared absorption spectra (ATR-SEIRAS). Figure 5e shows that, as the bias increases to 1.4 V vs. RHE, in-situ ATR-SEIRAS spectra of Ru exhibit absorption bands around the vibration frequency of 1038 cm$^{-1}$, corresponding to the *OOH/*OO intermediates. Simultaneously, a distinct absorption signal at the vibration frequency of 1189 cm$^{-1}$ gradually strengthens with the application of OER potential, corresponding to *O-O* in the LOM. In-situ ATR-SEIRAS spectra of Ru/Ti$_4$O$_7$ appear at similar positions as Ru, attributing to the absorption bands of *OOH/*OO (Fig. 5f). The broadening and shifting of absorption peaks may originate from the transition of the deprotonation process. Furthermore, almost no signal corresponding to *O-O* is observed around the vibration frequency of 1190 cm$^{-1}$. Additionally, the normalized density difference of in-situ ATR-SEIRAS spectra corresponding to *OOH/*OO and *O-O* signals is presented in Fig. 5g to determine the proportion occupied by adsorption evolution mechanism and LOM in the acidic OER process[38]. The higher adsorption density difference implies a higher proportion of adsorption evolution mechanism in the reaction process. In the high potential range, Ru undergoes surface reconstruction to form RuO$_x$ species, enabling both the adsorption evolution mechanism and LOM to drive OER. Meanwhile, in-situ ATR-SEIRAS spectra of Ru/Ti$_4$O$_7$ consistently exhibit a higher adsorption density difference than Ru during OER. This phenomenon indicates that Ru/Ti$_4$O$_7$ tends to drive OER through the adsorption evolution mechanism, which can realize more stable OER compared with the LOM with potential catalyst dissolution. In the pH-dependence tests, compared with Ru NPs, Ru/Ti$_4$O$_7$ showed less correlation with acidity, which verified that it was more inclined to drive OER with adsorption evolution mechanism (Supplementary Fig. 34). These results demonstrate that Ru/Ti$_4$O$_7$ tends to drive OER through the surface adsorption evolution mechanism, which can realize more stable OER compared with the LOM with potential catalyst dissolution[38,48].

Pulse voltammetry test was employed to assess the deprotonation capability of catalysts, which can confirm the source of the enhanced activity of Ru/Ti$_4$O$_7$ in OER[49,50]. Under different voltage pulses (Supplementary Fig. 35), Ru and Ru/Ti$_4$O$_7$ exhibit alternating cathodic and anodic current pulses (Fig. 5h). The oxidation charge storage capacity of different catalysts was further measured by integrating the anodic current response to voltage pulses. As shown in Fig. 5i, Ru/Ti$_4$O$_7$ demonstrates a higher oxidation charge storage capacity compared to Ru NPs, implying that Ru/Ti$_4$O$_7$ undergoes a faster deprotonation process to form reaction intermediate *O[49]. These findings suggest that the appropriate metal-support interactions between Ti$_4$O$_7$ and Ru can activate Ru sites by promoting the deprotonation process in OER. Furthermore, in the EIS bode plots, the pre-OER process of Ru/Ti$_4$O$_7$ results in an uneven distribution of surface charges (Supplementary Fig. 36), which is manifested by a reduction in the frequency peaks within the range of 1.40-1.45 V vs. RHE and a shift towards higher frequencies compared to the broader transition phase peaks of Ru NPs

(1.35–1.45 V vs. RHE). This phenomenon suggests that Ru/Ti$_4$O$_7$ exhibits a faster charge dissipation to accelerate deprotonation during OER to reaction kinetics[51]. In summary, the moderate metal-support interactions between Ru NPs and Ti$_4$O$_7$ through Ti–O–Ru units can reduce interface Schottky barriers to accelerate electron transfer and achieve electron enrichment at Ru sites to slow down the corrosion of Ru during OER[52–54]. In addition, electronic modulation of active Ru sites facilitates the deprotonation process of OER (Fig. 5j). During the HER process, EIS Nyquist plots at different current densities were recorded and fitted by the equivalent circuit inset of Supplementary Fig. 37a. $R_2$ reflects the hydrogen adsorption resistance on the material surface in the equivalent circuit[13,44]. Hydrogen adsorption kinetics were quantified by plotting log$R_2$ versus overpotential and calculating the EIS-derived Tafel slopes. As shown in Supplementary Fig. 37, the reduced slope of Ru/Ti$_4$O$_7$ of 21.3 mv dec$^{-1}$ compared with that of Ru/TiO$_2$ and Ru/TiO indicates the accelerated hydrogen adsorption kinetics, which is related to the hydrogen spillover (Supplementary Fig. 38, and Supplementary Table 4). The above results further demonstrate that Ru/Ti$_4$O$_7$ achieves fast HER kinetics through a potential hydrogen spillover mechanism.

## Theoretical calculation for intrinsic activity and stability analysis

Density functional theory (DFT) calculations reveal the impact of the support adaptability of Ti$_4$O$_7$ on the electronic structure modulation and the bifunctional reactivity of the Ru sites. The energy band structure manifests that Ti$_4$O$_7$ and TiO have a rich density of state distribution at the Fermi level, symbolizing their superior conductivity compared to a noticeable band gap of TiO$_2$ (Supplementary Fig. 39). The differential charge density shows a significant yellow area near the Ru atoms at the interface between Ru and Ti$_4$O$_7$, which verifies the electron enrichment of Ru atoms (Supplementary Figs. 40 and 36c)[55]. In the average potential profile (Supplementary Fig. 41d), the electrostatic potential difference between Ru and Ti$_4$O$_7$ can be more explicitly observed. Subsequently, Pourbaix diagrams were utilized to verify the stabilization of Ru based on Ti–O–Ru units between Ru and titanium oxide supports. According to the Pourbaix diagram of Ru-O (Fig. 6a), Ru is prone to over-oxidation to form high-valent Ru$^{4+}$ at high potentials for OER, especially in acidic environments. Furthermore, the robust interface Ti–O–Ru units effectively increases the theoretical dissolution potential of Ru in electrochemical oxidation as shown in the Pourbaix system of Ti–O–Ru (Fig. 6b). In Fig. 6c, the Bade and valence state calculations at the interface show that the Ru sites at the Ru/Ti$_4$O$_7$ interface have the highest valence state (−0.007), which is closest to the initial valence state of Ru and consistent with the XAFS results. This electronic structure adaptability gives Ru/Ti$_4$O$_7$ a lower interface Schottky barrier compared to Ru/TiO$_2$ and Ru/TiO to accelerate interface electron migration and trigger the hydrogen spillover mechanism for HER[47].

The structural model corresponding to the partially oxidized Ru NPs loaded on Ti$_4$O$_7$ was constructed to calculate the OER pathways based on the adsorption evolution mechanism (Supplementary Figs. 41–43). The local oxidation combined with the electron redistribution between Ru and different titanium oxide supports endow the Ru sites with varying intrinsic activities through the $d$-band center modulation. The downward shift of the $d$-band center reduces the energy of the antibonding orbitals formed by the adsorption reaction intermediates and the $d$-orbitals of Ru, which implies the weak adsorption of reaction intermediates[48,56]. Therefore, in the surface oxidized Ru/TiO, Ru/TiO$_2$ and Ru/Ti$_4$O$_7$, the $d$-band centers of the Ru sites are −1.80 eV, −1.81 eV and −2.07 eV, respectively (Supplementary Fig. 44 and Supplementary Table 5), which endow the corresponding *OH adsorption free energies with −0.482 eV, 0.435 eV, and 0.703 eV at U = 0 V, respectively (Fig. 6d). Subsequently, the deprotonation process is also optimized. Ru/Ti$_4$O$_7$ exhibits the smallest deprotonation

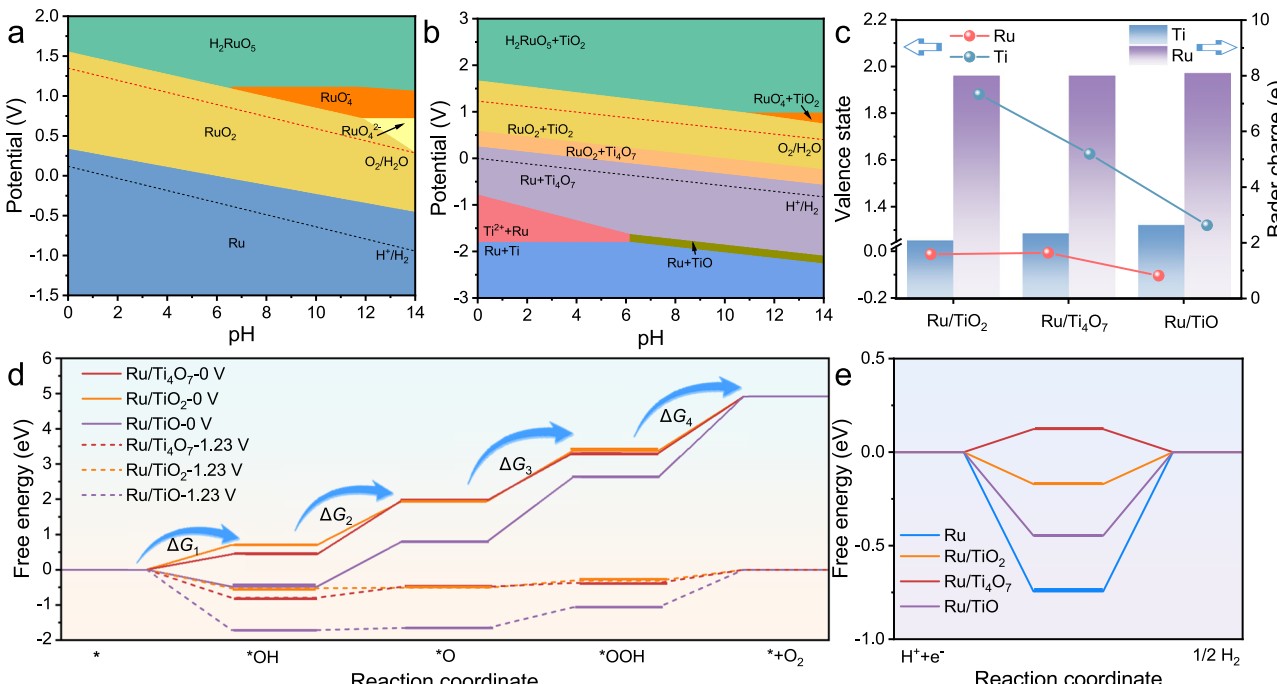

**Fig. 6 | Theoretical calculation and structure-activity relationship. a** Ru–O, and (**b**) Ti–O–Ru Pourbaix diagrams generated with aqueous ions concentration of $10^{-4}$ M at 25 °C. **c** Bade charge at the interface of Ru/TiO$_2$, Ru/Ti$_4$O$_7$, and Ru/TiO and corresponding calculated valence states of Ru and Ti. **d** Free energy profiles of different OER intermediates at 0 V and 1.23 V for Ru site in Ru, Ru/TiO$_2$, Ru/Ti$_4$O$_7$, and Ru/TiO. **e** Free energy profiles of HER intermediates for Ru site in Ru, Ru/TiO$_2$, Ru/Ti$_4$O$_7$, and Ru/TiO.

free energy and the lowest theoretical potential for OER ($\Delta G = 1.540$ eV, Supplementary Table 6). The enhanced deprotonation process is consistent with experimental results. The catalysts drive HER in its initial structure at the reduction potentials (Supplementary Fig. 45)[21,57]. The free energy step diagram verifies that the weak HER activity of Ru sites in Ru NPs originated from the strong adsorption of HER intermediates (Fig. 6e). Specifically, the *d*-band center of Ru at the interface exhibits distinct variations with the interface electron migration induced by the contact between Ru and different titanium oxides. The *d*-band center of Ru is −1.72 eV. After contact with TiO, TiO$_2$, and Ti$_4$O$_7$, the corresponding *d*-band centers of Ru shift to −1.77 eV, −1.89 eV, and −1.98 eV, respectively (Supplementary Fig. 46). Owing to the downward shift compared to the original Ru NPs induced by the metal-support interaction, the Ru sites in Ru, Ru/TiO, Ru/TiO$_2$, and Ru/Ti$_4$O$_7$ have *H adsorption free energies of −0.745 eV, −0.446 eV, −0.171 eV, and 0.123 eV, respectively (Fig. 6e, Supplementary Table 7)[58,59]. Therefore, Ru/Ti4O7 has the lowest theoretical reaction potential of 0.123 eV for HER. The theoretical calculations further verified that the appropriate support can promote the intrinsic activity of the Ru sites for both OER and HER.

### Performance of the water electrolysis device
Firstly, the unique advantages of suitable support design were verified through membrane-free water electrolysis tests. As shown in Fig. 7a, the water electrolysis driven by Ru/Ti$_4$O$_7$ with excellent bifunctional activity only requires a cell voltage of only 1.44 V to reach a current density of 10 mA cm$^{-2}$ with stable operation for 300 h (Fig. 7b) in the acidic environments. Noteworthily, the corresponding Faraday efficiency is also close to 100%. This excellent water splitting performance exceeds that of the state-of-the-art difunctional catalysts recently reported in the acidic media (Fig. 7c). The superiority of the Ti$_4$O$_7$ support for water splitting in the pH-universal environments has also been verified (Supplementary Figs. 47, 48 and Supplementary Table 8). Benefitting from the advantages of conductivity and stability, Ru/Ti$_4$O$_7$ was used as both cathode and anode catalysts in the MEA electrolyzers

(Fig. 7d). For the PEM water electrolysis (Fig. 7e), Ru/Ti$_4$O$_7$ ‖ Ru/Ti$_4$O$_7$ exhibits significantly lower cell voltage than that of RuO$_2$ ‖ Pt/C and maintains long-term operation of 200 h at 200 mA cm$^{-2}$ and 300 h at 500 mA cm$^{-2}$ (Fig. 7f and Supplementary Fig. 49), which is superior to most recently reported Ru-based catalysts (Supplementary Table 9). The unique structural advantages of the PEM electrolyzer and the characteristics of high-temperature operation combined with the excellent stability of Ru/Ti$_4$O$_7$ achieve robust PEM water electrolysis[60]. The EIS Nyquist curves of the MEA further verified the conductivity advantage of Ru/Ti$_4$O$_7$, which significantly reduces the energy consumption of the electrolyzer under high current densities (Supplementary Fig. 50)[61]. Furthermore, Ru/Ti$_4$O$_7$ also possesses extensible high activity and stability in AEM (Supplementary Fig. 51). The apparent advantages of bifunctional Ru/Ti$_4$O$_7$ in membrane electrode assembly show impressive industrialized application prospects.

## Discussion
In this study, titanium oxide designed through rational non-stoichiometric engineering was used to stabilize metal Ru for highly active and stable bifunctional water splitting. The non-stoichiometric design of Ti$_4$O$_7$ facilitates the achievement of appropriate metal-support interaction with Ru. The electron enrichment of Ru NPs through stable Ti–O–Ru units enhances the corrosion resistance of Ru to obtain partially active RuO$_x$ species, which inhibits the LOM and accelerates deprotonation for OER. The high electrical conductivity of Ti$_4$O$_7$ allows Ru/Ti$_4$O$_7$ to drive OER with low energy consumption. Furthermore, the work function adaptation between Ti$_4$O$_7$ and Ru makes HER tend to the hydrogen spillover mechanism to achieve fast reaction kinetics. As expected, Ru/Ti$_4$O$_7$ displays ultralow overpotentials of 8 mV and 150 mV at 10 mA cm$^{-2}$ for HER and OER in acidic environments, respectively, with steady operation for 500 h. The excellent difunctional performance enables Ru/Ti$_4$O$_7$ to drive water splitting at 10 mA cm$^{-2}$ at a voltage of 1.44 V, which is also validated in pH-universal environments. PEM devices assembled with Ru/Ti$_4$O$_7$ show lower cell voltages and higher stability than those of commercial

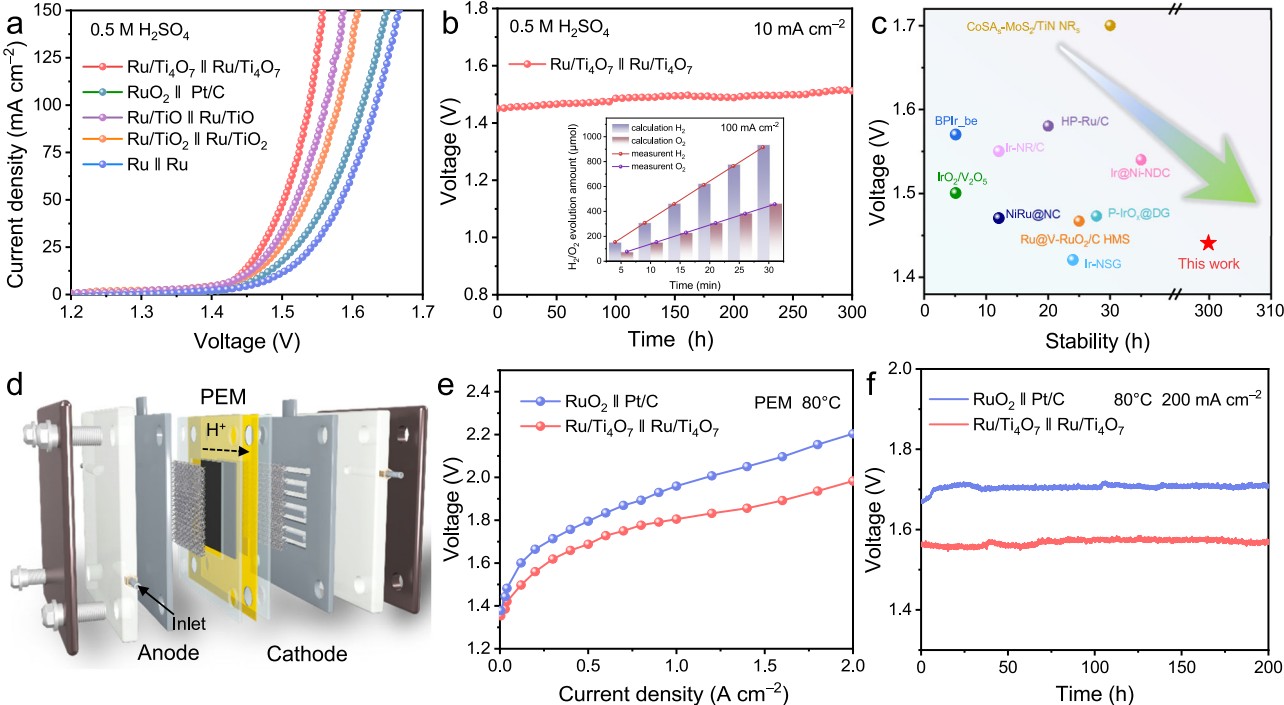

**Fig. 7 | The performance of water splitting. a** LSV polarization curves of Ru ‖ Ru, RuO$_2$ ‖ Pt/C, Ru/TiO$_2$ ‖ Ru/TiO$_2$, Ru/Ti$_4$O$_7$ ‖ Ru/Ti$_4$O$_7$, and Ru/TiO ‖ Ru/TiO (loading amount: 1 mg cm$^{-2}$) in 0.5 M H$_2$SO$_4$ (pH = 0.3) for the acidic water splitting. **b** Stability tests of Ru/Ti$_4$O$_7$ ‖ Ru/Ti$_4$O$_7$ at 10 mA cm$^{-2}$, and the inset display the corresponding Faraday efficiency at 100 mA cm$^{-2}$. **c** Comparison of the cell voltage and stability of Ru/Ti$_4$O$_7$ ‖ Ru/Ti$_4$O$_7$ with those of recently reported bifunctional catalysts for acidic water splitting. **d** Conceptual model of the PEM water electrolyzer flow cell using pure water as feedstock. **e** Steady polarization curves of Ru/Ti$_4$O$_7$ ‖ Ru/Ti$_4$O$_7$ and RuO$_2$ ‖ Pt/C (loading amount: 2 mg cm$^{-2}$) for PEM water electrolyzer in pure water (pH = 6.7) at 80 °C. **f** Stability tests of Ru/Ti$_4$O$_7$ ‖ Ru/Ti$_4$O$_7$ for PEM water electrolyzer at 200 mA cm$^{-2}$.

RuO$_2$ ‖ Pt/C and robust operation at 500 mA cm$^{-2}$ for 300 h. This outstanding performance validates the advancement of delicate support design for water electrolysis devices and opens new avenues for the rational construction of OER catalysts.

## Methods
### Material synthesis
**Preparation of Ti$_4$O$_7$.** Firstly, 1 g TiO$_2$ and 40 mg carbon black were ball-milled for 10 h to form a uniform mixture. Subsequently, the mixture was annealed at 1150 °C for 3.5 h to obtain Ti$_4$O$_7$ powder.

**Preparation of Ru/Ti$_4$O$_7$.** Ru-loaded Ti$_4$O$_7$ was prepared by simple wet reduction. First, 60 mg Ti$_4$O$_7$ and RuCl$_3$ were evenly dispersed in 50 mL of deionized water. Subsequently, 7 mL of 1 M NaBH$_4$ solution was slowly added dropwise into the mixture with vigorous stirring. After stirring for 4 h, the product was collected by centrifugation and dried under vacuum at 60 °C for 12 h. The obtained powder was further annealed at 300 °C, 400 °C, and 500 °C for 1 h in the Ar atmosphere. The product was labeled Ru/Ti$_4$O$_7$ and stored in the Ar-filled environment. The mass of Ti$_4$O$_7$ was maintained at 60 mg, and the dosage of RuCl$_3$ was changed to 20 mg, 40 mg, and 60 mg to achieve different Ru loading.

**Preparation of Ru/TiO$_2$ and Ru/TiO.** The preparation and storage methods of Ru/TiO$_2$ and Ru/TiO were the same as those of Ru/Ti$_4$O$_7$. TiO$_2$ and TiO replaced Ti$_4$O$_7$ to obtain Ru/TiO$_2$ and Ru/TiO, respectively.

### Material characterizations
XRD tests were carried out utilizing the Bruker D8 advance (Billerica) with a guaranteed scanning rate of 10° min$^{-1}$. SEM images were captured using the Regulus 8100 (Hitachi). AC-TEM (JEM-ARM200F, JEOL)

was employed to investigate the fine structure of the materials. The TEM and HRTEM images were acquired using the Tecnai G2 F20 (FEI). XPS utilized Al Kα X-ray as the excitation source (Escalab 250Xi, Thermo Fisher Scientific). All narrow XPS spectra were obtained under the conditions of 20 eV pass energy and 0.05 eV energy step, calibrated by the peak of C 1$s$ located at 284.8 eV. UPS was performed using a VG Scienta R4000 analyzer (monochromatic He I light source of 20.2 eV) with a 10 eV bias. XAFS measurements were conducted at the Taiwan Photon Source 44 A beamline quick-scanning X-ray absorption spectroscopy (Hsinchu). The XAFS spectra were collected at room temperature and analyzed using the Athena program. Inductively coupled plasma-mass spectrometry (ICP-MS) was carried out using the Optima 7300 DV instrument (PerkinElmer).

### Electrochemical measurements
The electrochemical tests were executed using an electrochemical workstation (Autolab PGSTAT302, Metrohm). The potentials were referenced to a reversible hydrogen electrode (RHE). The electrochemical measurements were performed in a three-electrode electrochemical cell with Ar-saturated 0.5 H$_2$SO$_4$, 1 M KOH aqueous solution, and 1 M phosphate buffered solution (PBS). The active catalysts and binder (Polyvinylidene fluoride, PVDF) were mixed in a weight ratio of 7:1 with N-methyl-2-pyrrolidone (NMP) as the solvent. The catalyst ink was then drop-dried onto the carbon paper and dried for preparation of the working electrode until the catalyst loading was 1 mg cm$^{-2}$. Commercial Pt/C (20 wt. %) and RuO$_2$ were prepared as the working electrodes using the same method for comparison. A platinum foil and a carbon rod were used as the counter electrodes for OER and HER, respectively. The Hg/Hg$_2$SO$_4$, Hg/HgO, and Ag/AgCl electrodes were employed as the reference electrode in acidic, alkaline, and neutral media, respectively. The CV tests at different scan rates were acquired within the 0.6−0.7 V vs. RHE range to determine

the $C_{dl}$. LSV curves were generated at a scan rate of 5 mV s⁻¹ with 85% $iR$ compensation. EIS was performed at the potentials of 1.5 V vs. RHE for OER and −0.5 V for HER, covering frequencies from 100 kHz to 0.1 Hz with an amplitude of 10 mV. In the pH-dependence measurement of OER, the electrolyte was prepared by adding the components of Britton-Robinson buffer (0.4 M each of phosphate, borate, and acetate) to a 0.5 M $Na_2SO_4$ solution, and the pH was then adjusted to the desired value by addition of $H_2SO_4$. All glassware was sonicated in ultrapure water directly before electrochemical treatment. Pulse voltammetry was performed while following the current over time. The potential was kept at a low potential ($E_l$ = 1.25 V vs. RHE), then switched and kept at a higher potential ($E_h$) before returning to $E_l$. This cycle was repeated while increasing $E_h$ from 1.42 V to 1.50 V in 20 mV/step and keeping $E_l$ unchanged. Charges related to the potential step were calculated by integrating the current pulse over time, accounting for the background current signal.

### In-situ electrochemical characterizations

**In-situ XAFS Measurements.** In-situ X-ray absorption spectroscopy, encompassing XANES and EXAFS at both Ru $K$-edge and Ti $K$-edge, was gathered in total-fluorescence-yield mode utilizing a silicon drift detector at the National Synchrotron Radiation Research Center (NSRRC) in Japan and Taiwan. The measurement, under the same conditions as the electrochemical characterization case in a typical three-electrode setup, took place in a specially designed Teflon container with a window sealed by Kempton tape. The scan ranges were kept in an energy range of 22,000–22,400 eV and 4900–5200 eV for Ru $K$-edge (BL-12B2 at SPring-8, NSRRC) and Ti $K$-edge (17C at Taiwan Light Source, NSRRC), respectively. The spectra were obtained by subtracting the baseline of the pre-edge and normalizing that of the post-edge. Fourier transform on $k^2$-weighted EXAFS oscillations was employed for EXAFS analysis. All EXAFS spectra are presented without phase correction.

**In-situ ATR-SEIRAS Measurements.** ATR-SEIRAS measurements were performed by a Nicolet iS50 Fourier transform infrared spectrometer (FT-IR) spectrometer with a liquid nitrogen-cooled MCT detector and a fixed angle IR optical path. The spectral resolution of the measurements was 8 cm⁻¹, and 32 interferograms were added for each spectrum.

### Membrane Electrode Assembly (MEA) measurements

**PEM electrolyzer measurements.** The membrane electrode assembly was prepared using Nafion 117 by the catalyst-coated membrane method with a geometric area of 1 cm × 1 cm. The bifunctional active catalysts were dispersed in isopropanol, deionized water, and a Nafion ethanol solution (5 wt%) to form an ink. Then, the ink was sprayed on both sides of PEM by the polytetrafluoroethylene (PTFE) transfer with an effective area of 1 × 1 cm² until the catalyst loading was 2 mg cm⁻². To avoid corrosion at oxidation potential, Ti felt was used as a gas diffusion layer (GDL) for the anode. Finally, the membrane with electrocatalysts coated, the anode GDL, and the cathode GDL (carbon paper) were hot pressed together to establish the MEA under 130 °C with a pressure of 10 MPa for 5 min. The commercial Pt/C (40 wt%) and $RuO_2$ were used in the same way as the cathode and anode respectively for comparison. Pure water was sent to the cathode and anode by a peristaltic pump at a speed of 40 rpm. Before testing, the prepared MEA was activated in potentiostatic mode at a cell voltage of 2 V. Subsequently, the electrolyzer was operated at 80 °C, and the hydrogen and oxygen in the electrolyte were removed with Ar to avoid the possible influence of the bubbles formed on the long-term stability of the electrodes. Steady-state polarization curves were obtained by a direct-current power (ITECH, IT-M3223). The cell voltages at different current densities were recorded.

**AEM electrolyzer measurements.** PiperION-A60 was utilized to construct the AEM electrolyzer. The catalyst ink of AEM was the same as that of PEM, except the Nafion ethanol solution was replaced by PiperION A ionomer ethanol (5 wt%). The ink was supported on carbon paper, and Ti felt with the effective area of 1 × 1 cm² until the catalyst loading was 2 mg cm⁻². To improve the interfacial contact between the catalyst layer and AEM, a small amount of PiperION A ionomer ethanol solution was sprayed on the surface of the anode and cathode catalyst layers and then dried at room temperature for 48 h. Finally, the AEM and the GDLs with catalyst loading were hot pressed together to establish the MEA under 100 °C with a pressure of 10 MPa for 5 min. The commercial Pt/C (40 wt%) and $RuO_2$ were also used in the same way as the cathode and anode respectively for comparison. One M KOH aqueous solution was sent to the cathode and anode by a peristaltic pump at a speed of 40 rpm. The electrolyzer was operated at 80 °C.

### Computational methods

All DFT calculations were conducted through the Vienna Ab initio Simulation Package (VASP). The computations utilized the projector augmented wave (PAW)[62] pseudopotential with the PBE[63] generalized gradient approximation (GGA) exchange-correlation function. The plane wave basis set had a cutoff energy of 500 eV. K-sampling in the calculation of adsorption energy used a Monkhorst-Pack mesh of 1 × 1 × 1, while in the calculation of DOS, a 5 × 5 × 1 mesh was employed. Spin polarization was applied to all structures, and complete relaxation of all atoms was ensured with an energy convergence tolerance of 10⁻⁵ eV per atom. The final force on each atom was maintained below 0.05 eV Å⁻¹. Porbaix diagrams were calculated using atomic simulation environment (ASE)[64] with input formation energy by DFT calculations of bulk and surface models.

In the construction of heterojunctions, the minimum lattice mismatch is used as a criterion to construct the interface contact model between Ru and different titanium oxides. The (0 0 1) facet of TiO contacts the (0 1 0) facet of Ru. The (1 − 1 0) facet of $TiO_2$ contacts the (2 1 0) facet of Ru. The (1 1 − 1) facet of $Ti_4O_7$ contacts the (0 0 1) facet of Ru.

The adsorption energy of reaction intermediates can be computed using the following Eq. (1):

$$\Delta G_{ads} = E_{ads} - E * + \Delta E_{ZPE} - T\Delta S \tag{1}$$

Where ads = (*H, *OH, *O, *OOH), and ($E_{ads} - E_*$) is the binding energy, $\Delta E_{ZPE}$ is the zero-point energy change, $\Delta S$ is the entropy change. In this work, the values of $\Delta E_{ZPE}$ and $\Delta S$ were obtained by vibration frequency calculation.

The Gibbs free energy of the five reaction steps can be calculated by the following Eqs. (2–7):

$$* + H_2O = *OH + H^+ + e^- \tag{2}$$

$$*OH = *O + H^+ + e^- \tag{3}$$

$$*O + H_2O = *OOH + H^+ + e^- \tag{4}$$

$$*OOH = * + O_2 + H^+ + e^- \tag{5}$$

$$* + H^+ + e^- = *H \tag{6}$$

$$*H = * + 1/2H_2 \tag{7}$$

In this work, $\Delta G_{1-4}$ were calculated at U = 0.

## Data availability

The data generated in this study are provided in the Source Data file.

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

## Acknowledgements

This work was supported by the National Natural Science Foundation of China (22075141), Scientific and Technological Innovation Special Fund for Carbon Peak and Carbon Neutrality of Jiangsu Province (BK20220039), and Natural Science Foundation of Jiangsu Province (BK20210311). The authors thank the National Synchrotron Radiation Research Center (NSRRC), Hsinchu, Taiwan.

## Author contributions

S.Z. performed the experiments, collected the data, and analyzed the data. S.Z. and L.D. wrote the manuscript. T.X. performed the experiments. S.H. and W.Z. performed the in-situ XAFS test. S.L. and F.H. analyzed the data and wrote the manuscript. C.K. and H.C. performed the XAFS measurement. S.P. designed the study, analyzed the data, and revised the manuscript.

## Competing interests

The authors declare no competing interests.
