## [Peer Review File · Nature Communications]

Reviewers' comments:

Reviewer #1 (Remarks to the Author):

The authors synthesized defect regulation of Ti₄O₇ with high conductivity and chemical stability endows the supports and Ru nanoparticles with HER and OER performance in pH-universal environments. The authors consider that interface crystal structure matching between Ti₄O₇ and Ru forms Ti–O–Ru units to stabilize Ru species during OER and promote electron transport. Hydrogen spillover induced by the low work function difference between Ti₄O₇ and Ru accelerates the reaction kinetics of HER. In acidic media, Ru/Ti₄O₇ displays overpotentials of 8 mV and 150 mV for HER and OER with operation of 500 hours at 10 mA cm⁻². However, there are some concerns regarding the novelty of work in terms of materials and mechanisms. Ru-based electrocatalytic materials for HER and OER have been widely reported (New J. Chem., 2023,47, 9628-9634; Inorg. Chem. 2022, 61, 48, 19407–19416), and the novelty of this work has not been discovered. This manuscript is not conclusive or novel enough to provide the needed new momentum for mechanistic discussion in the field. The stability test of 500 hours seems to be considerable, but this was performed at a current density of 10 mA cm⁻², which obviously casts deep doubt on the application. Including the assembled water electrolysis device, there is still no obvious advantage in performance compared with what has been reported. What is even more regrettable is that no detection or discussion of the key intermediate reactants of OER has been found in the entire manuscript, leaving important questions about the OER mechanism unclear. The Fourier transform r-space assignment for Ru/Ti₄O₇ is weak and the evidence presented is weaker than claimed. On balance, it is recommended to reject the current work.

Reviewer #2 (Remarks to the Author):

Peng et al. reported the construction of regulable supports via non-stoichiometric engineering to stabilize metal ruthenium for enhanced pH universal water-splitting. The Ru/Ti₄O₇ composite showed excellent performance for both the Hydrogen Evolution Reaction (HER) and Oxygen Evolution Reaction (OER) in pH-universal environments. Density Functional Theory (DFT) calculations were performed to provide insight into the origin of the good performance. Various types of calculations were carried out, including Pourbaix diagrams, band structures, density of states, and reaction profiles. However, it is still not clear how these calculations relate to the adsorption strength of different intermediates and, ultimately, the performance. More detailed analysis and computational details are required for a clear understanding.

Importantly, I have stronger concerns about the chemical structure models employed to study the reaction mechanisms. Since it is an interface model, which surfaces of TiO₂, Ti₄O₇, and TiO were used to

contact with what surface of Ru? This information is crucial to know if the interfaces have a proper lattice match and if the right surfaces were used with respect to the experimental ones. Additionally, how did the authors determine the most stable models? The author used a sandwich interface model (Fig 30c in SI) and claimed charge accumulation on Ru atoms, connecting it with the performance. However, this contact is different from the interface model (FigS 31-33 in SI) for HER and OER analysis. These two different models have different exposed sites with different chemical environments. Consequently, it is not suitable to be used together to explain the good performance.

The surface states of Ru in FigS 31-33 in SI are different, creating a new factor that affects the adsorption strength of *O, *OH, and *OOH. This negatively affects the arguments on the lower overpotential due to the Ru/Ti4O7 interface.

The author claimed that electron migration between Ru and titanium oxide forms electron-rich Ru sites and weakens the adsorption of the HER intermediates. This is a very general comment for the Ru/TiO2, Ru/Ti4O7, and Ru/TiO, leaving the question of why Ru/Ti4O7 had better performance unanswered.

Reviewer #3 (Remarks to the Author):

The authors have developed a Ti4O7-loading Ru nanoparticles catalyst for pH-universal water-splitting. Ru/Ti4O7 exhibits ultralow overpotentials of 8 mV and 150 mV for HER and OER, respectively, maintaining stable performance over 500 hours at 10 mA cm⁻² in acidic media.

I thank the authors for demonstrating the catalyst's outstanding activities. However, the stability concerns raised in the manuscript need to be addressed, major revisions are recommended before being considered for publication in Nature Communications.

1. In the manuscript (line 93), the authors state, "The suitable oxygen vacancies in Ti4O7 promote the Ru³⁺ adsorption on Ti4O7, which endows the Ru NPs with a more uniform distribution on Ti4O7 compared to TiO2 with low vacancies." I am puzzled by whether Ru can attach to oxygen vacancies when there is no oxygen present in these vacancies. How does Ru bond to Ti in the absence of oxygen?

2. Ti4O7 possesses oxygen vacancies, can these vacancies remain stable under high-current OER conditions? In extensive oxygen-deficient materials, defects can be filled with reactive oxygen during the OER process. Can the structure of Ti4O7 withstand this, especially considering the observed continuous leaching of basal Ti in Figure 4f? Authors are encouraged to provide evidence.

3. The authors emphasize "Interface crystal structure matching between Ti₄O₇ and Ru forms Ti–O–Ru units to stabilize Ru species during OER." Why was stability not measured under high-current acidic conditions, and only at 10 mA/cm²? The stability decay rate at 10 mA/cm² (may 142 μV/h), calculated from Figure 3h and Figure 6b, can definitely not demonstrate its stability.

4. It is recommended that the authors supplement the manuscript with CV after the OER stability test, along with ICP of Ru in the catalyst and XPS data. In Figure 4f, for the Ru solution ICP, a platform was not observed but an increase, and it is suggested that the authors extend the testing time to 100 hours.

5. I am curious about the reason behind the bipolar stability being lower than MEA stability, especially when the stability at 10 mA/cm² is evidently decaying in the article. Why does the stability improve in MEA?

6. In Figure 5b, it is noticeable that Ru is more prone to dissolution in a pH=14 environment. Why does stability in an AEM (alkaline environment) (500 mA/cm²) surpass that in a PEM (200 mA/cm²)?

7. In Line 142, "reduced valence" possibly an error, should it be "oxide valence"?

8. In Line 173, SI Table 1 is mentioned as mass activity assemble, but it is actually ICP-MS data.

Manuscript ID: NCOMMS-23-53716
Point-by-Point Response to Review Comments

We sincerely appreciate the pertinent suggestions of editors and reviewers. The quality of the manuscript has been significantly improved following reviewers' comments. All modifications have been highlighted with yellow shading in the revised manuscript.

Reviewers' comments:

Reviewer #1:

Comments:

The authors synthesized defect regulation of Ti_4O_7 with high conductivity and chemical stability, which endows the supports and Ru nanoparticles with HER and OER performance in pH-universal environments. The authors consider that interface crystal structure matching between Ti_4O_7 and Ru forms Ti–O–Ru units to stabilize Ru species during OER and promote electron transport. Hydrogen spillover induced by the low work function difference between Ti_4O_7 and Ru accelerates the reaction kinetics of HER. In acidic media, Ru/ Ti_4O_7 displays overpotentials of 8 mV and 150 mV for HER and OER with operation of 500 hours at 10 mA cm^{-2} .

Our response: We express our gratitude for the reviewer's dedicated time and effort in providing feedback on our manuscript. In response to the corresponding comments, we have further emphasized the novelty of this work and provided in-depth insights into the reaction mechanism. We hope that our response can enable the reviewer to clearly understand the unique significance of our work for membrane electrode assembly (MEA) water electrolysis.

Comments:

However, there are some concerns regarding the novelty of work in terms of materials and mechanisms. Ru-based electrocatalytic materials for HER and OER have been widely

reported (New J. Chem., 2023,47, 9628-9634; Inorg. Chem. 2022, 61, 48, 19407–19416), and the novelty of this work has not been discovered. This manuscript is not conclusive or novel enough to provide the needed new momentum for mechanistic discussion in the field.

Our response: We thank the reviewer for the great efforts in improving our work. Based on the valuable suggestions, we carefully checked our work and made many revisions. We conducted a detailed comparison and analysis of the articles you cited. We further emphasized the innovation of this work, including the dedicated support design, excellent bifunctional activity of OER and HER, and in-depth analysis of the mechanism of the metal-support interaction.

As indicated by the literature [*New J. Chem.* 2023, 47, 9628], Li et al. used a dye-sensitized strategy to load hyperdispersed Ru nanoparticles on the TiO₂ surface. A carbon layer is a support to enhance electrical conductivity. The obtained C/Ru/TiO₂ has an overpotential of 69 mV at 10 mA cm⁻² for HER in 0.5 M H₂SO₄ with a Tafel slope of 70 mV dec⁻¹. In 1 M KOH, the HER overpotential is only 51 mV at 10 mA cm⁻², with a Tafel slope of 68 mV dec⁻¹. Furthermore, Anup Kuchipudi et al. have prepared a shape-selective LaCrO₃ and modified Ru nanoparticles via a hydrothermal method to synthesize RLCO₂. The lowest HER overpotential of the composite material RLCO₂ is 150 mV [*Inorg. Chem.* 2022, 61, 19407]. These works indicate that Ru species with various supports, such as TiO₂ and LaCrO₃, have sufficient potential for HER and OER. However, in these works, the poor intrinsic conductivity of the TiO₂ leads to low electrochemical activity. Meanwhile, the impact of the metal-support interaction mechanism on the reaction process and the actual active species has not been revealed. The catalysts with high activity and stability for both HER and OER still have not been developed in these works. However, our work mainly focuses on OER activity, which is the decisive reaction in PEMWE. Therefore, in-depth research on the related systems is necessary.

Our work addresses the shortcomings of the low conductivity support and shows sufficient innovation and essential contributions to the field in the following content with sufficient data:

(1) **Non-stoichiometric design of the supports.** Loading active species on suitable

support not only enhances the stability of the active species but also introduces interface active sites through appropriate metal-support interactions to enhance catalytic activity for OER and HER. However, common stable supports for electrocatalytic water electrolysis in acid, such as TiO₂, MnO₂, and WO₃, exhibit poor conductivity [Nat. Catal. 2021, 4, 1012-1023; Nat. Commun. 2022, 13, 5382; Angew. Chem. Int. Ed. 2023, 62, e202300406]. The low conductivity of supports leads to high operating voltages of electrocatalytic water devices, which increases energy consumption and accelerates the dissolution of active species [Angew. Chem. Int. Ed. 2023, 62, e202216645; Angew. Chem. Int. Ed. 2023, 62, e202308704; Adv. Energy Mater. 2020, 10, 1902521]. Therefore, the ideal supports need to possess both high conductivity and structural stability simultaneously. Defect engineering can effectively enhance the conductivity of semiconductors [Energy Environ. Sci. 2020, 13, 5143; Adv. Mater. 2023, 35, 2306097]. However, the random distribution of oxygen vacancies makes it challenging to precisely achieve defect state modulation and results in reduced crystallinity and potential structural damage to semiconductors [Appl. Catal. B Environ. 2022, 310, 121332.; Nano Energy 2020, 73, 104761]. Thus, non-stoichiometric ratio engineering was employed to finely customize titanium oxide supports (TiO, Ti₄O₇, and TiO₂) with oxygen vacancies of periodic arrangements. Ti₄O₇, with regularly arranged defects, emerges as a potential competitor with superior structural stability over multi-defect TiO and significantly higher conductivity than TiO₂ (Supplementary Fig. 38).

Supplementary Fig. 38 The energy band structures of **a** TiO₂, **b** Ti₄O₇, and **c** TiO, respectively. The energy band structure manifests that Ti₄O₇ and TiO have a rich density of state distribution at the Fermi level, symbolizing their superior conductivity compared to a noticeable band gap of TiO₂.

As shown in **Fig. 1**, non-stoichiometric ratio engineering was employed to finely customize titanium oxide supports (TiO, Ti₄O₇, and TiO₂) with oxygen vacancies of periodic arrangements. Ti₄O₇, with regularly arranged defects, emerges as a potential competitor with superior structural stability over multi-defect TiO and significantly higher conductivity than TiO₂. More importantly, Ti₄O₇ exhibits a suitable metal-support interaction, which lays the foundation for the dual-function activity of Ru/Ti₄O₇ in both HER and OER in acid and alkaline. On the one hand, the higher work function compared to Ru NPs promotes the electron richness of Ru, which can alleviate the dissolution of Ru in OER. On the other hand, compared to TiO and TiO₂, the minimum work function difference between Ti₄O₇ and Ru NPs ($\Delta\Phi = 0.30$ eV) can reduce the interface Schottky barrier, which further promotes electron transport in the composite catalyst and trigger hydrogen spillover during HER. The related details have been added in the revised manuscript, Page 5, Lines 85-94, highlighted in yellow.

Fig. 1 Schematic diagram of the interaction mechanism between different titanium oxide supports and Ru NPs to activate OER and HER.

(2) **Dual-functional activation and stability enhancement of OER and HER.** Ir-based materials have been recognized for the excellent stability in acidic environments [*Nat.*

Commun. 2023, 14, 4127; *Angew. Chem. Int. Ed.* 2023, 62, e202313954; *Angew. Chem.* 2023, 135, e202216645]. However, the low mass activity of Ir-based materials, coupled with high costs, severely impedes their large-scale practical applications. Ru, with its lower cost (approximately 1/6 of Ir) and higher intrinsic activity, stands as a more promising commercial candidate [*Nat. Commun.* 2023, 14, 5365; *Nat. Commun.* 2023, 14, 5365; *J. Am. Chem. Soc.* 2023, 145, 43, 23659]. Additionally, Ru exhibits significantly higher catalytic activity for HER compared to Ir, making it a potentially efficient dual-functional catalyst for PEM electrolyzers to reduce device costs further [*Nat. Mater.* 2023, 22, 100; *Nat. Commun.* 2022, 13, 5448; *Nat. Catal.* 2, 2019, 304]. Therefore, Ru-based catalysts have received widespread attention and show great application potential for water electrolysis. In-depth research on the reaction mechanism of Ru-based catalysts for OER and HER is also of great significance [*Nat. Catal.* 2021, 4, 711; *Nat. Catal.* 2021, 5, 212; *Nat. Synth.* 2023, doi: 10.1038/s44160-023-00444-x]. However, during the OER process, Ru forms soluble Ruⁿ⁺ species (n > 4) with oxidizing potentials in harsh reaction environments, leading to a short lifespan. Furthermore, the intrinsic activity of Ru is lower than that of Pt due to the strong adsorption energies of the reaction intermediates. Our work addresses these problems associated with Ru-based catalysts by constructing a stable non-stoichiometric support, Ti₄O₇. The appropriate metal-support interaction imparts Ru/Ti₄O₇ with superior activity and stability for both OER and HER compared to Ru/TiO and Ru/TiO₂. As shown in **Fig. R1**, Ru/Ti₄O₇ exhibits ultra-low overpotentials of 8 mV and 150 mV for HER and OER at 10 mA cm⁻², respectively, maintaining a long lifespan of 500 hours in acidic environments.

Fig. R1 **a** Linear sweep voltammetry (LSV) polarization curves of OER and **b** Tafel plots of Ru, commercial RuO₂, Ru/TiO₂, Ru/Ti₄O₇, and Ru/TiO in 0.5 M H₂SO₄. **c** LSV polarization curves of HER and **d** Tafel plots of Ru, commercial Pt/C, Ru/TiO₂, Ru/Ti₄O₇, and Ru/TiO in 0.5 M H₂SO₄. **e** Chronopotentiometry curves of Ru/Ti₄O₇ at 10 mA cm⁻², and -10 mA cm⁻² in 0.5 M H₂SO₄.

Furthermore, at a high current density of 200 mA cm⁻², Ru/Ti₄O₇ maintains stable operation for 300 hours with only a performance decay of 0.12 mV h⁻¹ (**Supplementary Fig. 24**). The PEM device assembled with Ru/Ti₄O₇ as both cathode and anode catalysts can continuously operate for 300 hours at 500 mA cm⁻², demonstrating unique advantages among recently reported Ru catalysts and showing potential industrial applicability (**Supplementary R2**). The related details have been added in the revised manuscript, Page 10, Lines 188-190; Page 19, Lines 369-370, highlighted in yellow.

Supplementary Fig. 24 Chronopotentiometry curves of Ru/Ti₄O₇ at 200 mA cm⁻² in 0.5 M H₂SO₄.

Supplementary R2 a Conceptual model of the PEM water electrolyzer flow cell using pure water as feedstock. **b** Stability tests of Ru/Ti₄O₇ || Ru/Ti₄O₇ for PEM water electrolyzer at 500 mA cm⁻².

(3) **Detailed reaction process monitoring and in-depth mechanistic analysis.** In acidic OER, the active sites primarily follow the adsorbed evolution mechanism (AEM) and lattice oxygen mechanism (LOM). Compared to AEM, LOM exhibits higher oxygen evolution activity but leads to a decrease in stability. Typically, OER involves both AEM and LOM simultaneously. The proportion of LOM is influenced by factors such as the crystallinity of the catalyst and defect concentration. Ru-based catalysts show high OER activity but poor stability, attributed to their elevated proportion of LOM during the OER

process. In this study, we observed OER intermediate species through *in-situ* attenuated total reflection-surface enhanced infrared absorption spectra (ATR-SEIRAS) to determine the specific reaction mechanism. *In-situ* ATR-SEIRAS provides direct evidence of the suitable metal-support interaction between Ti₄O₇ and Ru species, inhibiting LOM during the OER process. More importantly, the appropriate metal-support interaction alters the electronic state of the interface Ru species, endowing Ru/Ti₄O₇ with superior deprotonation ability compared to Ru/TiO₂ and Ru/TiO, effectively reducing the OER overpotential (**Fig. 5**). The related details have been added in the revised manuscript, Pages 13-14, Lines 251-270; Page 14-15, Lines 27-285, highlighted in yellow.

Fig. 5 *In situ* ATR-SEIRAS spectra of **e** Ru and **f** Ru/Ti₄O₇ for OER in 0.5 M H₂SO₄ under different potentials vs. RHE. **g** Intensity difference of the infrared signals at 1038 and 1189 cm⁻¹.

Comments:

The stability test of 500 hours seems to be considerable, but this was performed at a current density of 10 mA cm⁻², which obviously casts deep doubt on the application. Including the assembled water electrolysis device, there is still no obvious advantage in performance compared with what has been reported.

Our response: We thank the reviewer for the constructive comments. Overpotential and stability tests of OER and HER at the current density of 10 mA cm⁻² are currently the most common and widely recognized activity and stability evaluation indicators of the catalysts. Therefore, we first highlighted the advantages of electrocatalytic activity at 10 mA cm⁻². In this work, Ru/Ti₄O₇ exhibits excellent catalytic activity and stability for both OER and HER in acidic environments. Furthermore, in alkaline and neutral environments,

the performance of this material is also impressive. As demonstrated in **Supplementary Tables 2, 3, and 8**, the outstanding performance of Ru/Ti₄O₇ stands out among numerous electrocatalytic materials for water electrolysis reported in recent years.

Supplementary Table 2. Comparison of OER performance of this work with that of pH-universal catalysts recently reported.

Catalysts	Noble metal content	Electrocatalytic activity						Reference
		0.5 M H ₂ SO ₄		1 M KOH		1 M PBS		
		η_{10} (mV)	Stability@10 mA cm ⁻² (h)	η_{10} (mV)	Stability@10 mA cm ⁻² (h)	η_{10} (mV)	Stability@10 mA cm ⁻² (h)	
Ru/Ti ₄ O ₇	20.98 wt% Ru	150	500	180	300	220	300	This work
Ni-RuO ₂	15.52 wt% Ru	214	200	/	/	/	/	Nat. Mater. 2023, 22, 100.
12Ru/MnO ₂	11.6 wt% Ru	161	200	/	/	/	/	Nat. Catal. 2021, 4, 1012.
Ru-Pt ₃ Cu	0.0163 mg _{Pt+Ru} cm ⁻²	280	28	/	/	/	/	Nat. Catal. 2019, 2, 304.
0.2Mo-PIO	21.73 wt% Ir	295	200	/	/	/	/	Nat. Commun. 2023, 14, 4127.
AuSA-MnFeCoNiCu LDH	1.1 wt% Au	/	/	213	700 h@100 mA cm ⁻²	/	/	Nat. Commun. 2023, 14, 6019.

Ir-NSG	7.33 wt% Ir	265	1.25	256	1.7	297	4.2	Nat. Commun. 2020, 11, 4246.
RuIrZnO _x	20 wt% Ru+Ir	240	24	220	24	/	/	Nat. Commun. 2019, 10, 4875.
Ru/Co–N–C	0.36 wt% Ru	232	20	276	20	400	20	Adv. Mater. 2022, 34, 2110103.
Ru@V-RuO ₂ /C HMS	0.28 mg cm ⁻²	176	15	201	15	/	/	Adv. Mater. 2023, 35, 2206351.
FRNO/CC	0.527 mg cm ⁻²	200	100	260	100	/	/	Adv. Energy Mater. 2023, 13, 2300174.
RuCu NSs/C	/	236	15	234	20	/	/	Angew. Chem. Int. Ed. 2019, 58, 13983–13988.
Ni-cluster-Ru NWs	20 wt% Ru	205	10	194	10	/	/	Energy Environ. Sci., 2021, 14, 3194–3202.
BPIr _{be}	0.13 mg cm ⁻²	290	500	290	500	620	/	Adv. Mater. 2021, 33, 2104638.
IrO ₂ /V ₂ O ₅	19.70 wt% Ir	266	20	283	20	329	20	Adv. Sci. 2022, 9, 2104636.
Ir-COP	9.81 wt% Ir	242	36	230	36	/	/	Adv. Funct. Mater.

Ir@Ni-NDC	9.72 wt% Ir	219	/	210	/	296	/	2023, 33, 2211192. Angew. Chem. Int. Ed. 2023, 62, e202302220.
Pd/NiFeO _x	/	169	50	180	50	310	50	Adv. Funct. Mater. 2021, 31, 2107181.
a/c-RuO ₂	/	220	20 h@50 mA cm ⁻²	235	/	287	/	Angew. Chem. Int. Ed. 2021, 60, 18821– 18829.

Supplementary Table 3. Comparison of HER performance of this work with that of pH-universal catalysts recently reported.

Catalysts	Noble metal content	Electrocatalytic activity						Reference
		0.5 M H ₂ SO ₄		1 M KOH		1 M PBS		
		η_{10} (mV)	Stability@10 mA cm ⁻² (h)	η_{10} (mV)	Stability@10 mA cm ⁻² (h)	η_{10} (mV)	Stability@10 mA cm ⁻² (h)	
Ru/Ti ₄ O ₇	20.98 wt% Ru	8	500	14	300	27	300	This work
s-Pt/1T' - MoS ₂	12.20 wt% Pt	19	240 h@1500 mA cm ⁻²	/	/	/	/	Nature 2023, 621, 300.
Pt ₁ /OLC	0.27 wt% Pt	38	100 h@40 mA cm ⁻²	/	/	/	/	Nat. Energy 2019, 4, 512.
Li-Pd ₃ P ₂ S ₈	51.10 wt% Pd	91	13 h@20 mA cm ⁻²	/	/	/	/	Nat. Catal. 2018, 1, 460.
B-Os Aerogel	99.3 wt% Os	12	202	19	20	33	20	Nat. Commun. 2022, 13, 1143.

Ru@C ₂ N	0.285 mg cm ⁻²	22	/	17	/	/	/	Nat. Nanotech. 2017, 12, 441.
Ir-NSG	7.33 wt% Ir	17	11	18.5	17.7	16.8	1.7	Nat. Commun. 2020, 11, 4246.
RuIrZnO _x	20 wt% Ru+Ir	13	24	14	24	/	/	Nat. Commun. 2019, 10, 4875.
Pt ₃ Fe/NMCS-A	10.3 wt% Pt	13	10	29	10	48	10	Adv. Mater. 2023, 35, 2303030.
Ru/Co-N-C	0.36 wt% Ru	17	20	19	20	87	20	Adv. Mater. 2022, 34, 2110103.
Ru@V-RuO ₂ /C HMS	0.28 mg cm ⁻²	47	15	6	15	/	/	Adv. Mater. 2023, 35, 2206351.
RuCu NSs/C	/	20	/	19	/	/	/	Angew. Chem. Int. Ed. 2019, 58, 13983.
FRNO/CC	0.527 mg cm ⁻²	30	80	82	80	/	/	Adv. Energy Mater. 2023, 13, 2300174.
Ni-cluster-Ru NWs	20 wt% Ru	20	10	17	10	/	/	Energy Environ. Sci., 2021, 14, 3194–3202.
BPIr _{be}	0.13 mg cm ⁻²	26	100	1.98	100	329	/	Adv. Mater. 2021, 33, 2104638.
Ir-COP	9.81 wt% Ir	12.3	100	14.5	100	/	/	Adv. Funct. Mater. 2023, 33, 2211192.
Ir@Ni-NDC	9.72 wt% Ir	41	/	19	/	31	/	Angew. Chem. Int. Ed. 2023, 62, e202302220.
RuO ₂ -300Ar	0.2 mg cm ⁻²	16	/	17	300	29	/	Energy Environ. Sci. 2021, 14, 5433.
Ru@IT-MoS ₂ -MXene	15 wt% Ru	44	160	42	100	106	100	Adv. Funct. Mater. 2023, 33, 2212514.
Ru@Ni-MOF	2.3 wt% Ru	37	24 h @ 100 mA cm ⁻²	22	24 h @ 100 mA cm ⁻²	52	24 h @ 100 mA cm ⁻²	Angew. Chem. Int. Ed. 2021, 60, 22276–2282.

Supplementary Table 8. Comparison of the acidic water splitting performance of this work with that of the noble metal based catalysts recently reported.

Catalysts	Electrocatalytic activity						Reference
	0.5 M H ₂ SO ₄		1 M KOH		1 M PBS		
	Voltage @ 10 mA cm ⁻² (V)	Stability @ 10 mA cm ⁻² (h)	Voltage @ 10 mA cm ⁻² (V)	Stability @ 10 mA cm ⁻² (h)	Voltage @ 10 mA cm ⁻² (V)	Stability @ 10 mA cm ⁻² (h)	
Ru/Ti ₄ O ₇	1.44	300	1.44	300	1.53	300	This work
Ir-NSG	1.42	24	1.45	24	1.53	24	Nat. Commun. 2020, 11, 4246.
RuIrO _x	1.45	24	1.47	24	1.49	24	Nat. Commun. 2019, 10, 4875.
CPF-Fe/Ni	1.44	120	1.57	120	/	/	Nat. Commun. 2023, 14, 1792.
RuCu NSs/C	1.49	15	1.49	40	/	/	Angew. Chem. Int. Ed. 2019, 58, 13983–13988.
Ru/Co–N–C	1.49	/	1.5	/	/	/	Adv. Mater. 2022, 34, 2110103.
Ru@V- RuO ₂ /C HMS	1.47	25	1.44	25	1.47	25	Adv. Mater. 2023, 35, 2206351.
Ni-cluster- Ru NWs	1.45	/	1.44	/	/	/	Energy Environ. Sci. 2021, 14, 3194–3202.
BPIr _{be}	1.57	5	1.54	15	2.11	/	Adv. Mater. 2021, 33, 2104638.
IrO ₂ /V ₂ O ₅	1.5	30	1.49	30	1.65	30	Adv. Sci. 2022, 9, 2104636.
Ir@Ni-NDC	1.54	35	1.46	100	1.59	30	Angew. Chem. Int. Ed. 2023, 62, e202302220.
HP-Ru/C	1.58	20	1.61	20	1.61	20	Appl. Catal. B- Environ. 2021, 294, 120230.

Additionally, according to the reviewer's suggestions, we have further supplemented stability tests for acidic OER at high current density. We have validated the performance of the proton exchange membrane (PEM) assembled with Ru/Ti₄O₇ as both cathode and anode catalysts under high current density conditions. The experimental results indicate that Ru/Ti₄O₇ can maintain robust operation for 300 hours at a high current density of 200 mA cm⁻² due to the appropriate interface electron redistribution between Ru NPs and Ti₄O₇ (**Supplementary Fig. 24**). The PEM electrolyzer assembled with Ru/Ti₄O₇ demonstrates sustained operation for 300 hours at 500 mA cm⁻² (**Supplementary R2**). The performance exhibits a significant advantage over reported Ru-based PEM

electrolysis devices to date (**Supplementary Table 9**). Moreover, in the existing literature on PEM electrolysis, most catalytic materials can only catalyze either HER or OER in acidic environments. Since Ru/Ti₄O₇ exhibits excellent activity and stability for both OER and HER, the PEM electrolyzer assembled with Ru/Ti₄O₇ does not require the assistance of commercial precious metal catalysts (Pt/C, RuO₂, or IrO₂), which effectively reduces the cost of catalysts in PEM electrolyzers. The related details have been added in the revised manuscript, Page 10, Lines 188-190; Page 19, Lines 369-370, highlighted in yellow.

Supplementary Fig. 24 Chronopotentiometry curves of Ru/Ti₄O₇ at 200 mA cm⁻² in 0.5 M H₂SO₄.

Supplementary R2 a Conceptual model of the PEM water electrolyzer flow cell using pure water as feedstock. **b** Stability tests of Ru/Ti₄O₇ || Ru/Ti₄O₇ for PEM water electrolyzer at 500 mA cm⁻².

Supplementary Table 9. Comparison of the PEM performance of this work with that of the Ru based catalysts recently reported.

Catalysts	E (V)@J (A cm ⁻²)	Stability (h) @J (A cm ⁻²)	Reference
Ru/Ti₄O₇	1.69@0.5	300@0.5	This work
Ni-RuO ₂	1.78@0.5	1000@0.2	Nat. Mater. 2023, 22, 100.
SS Pt-RuO ₂ HNSs	\	100@0.1	Sci. Adv. 2022, 8, eab19271.
Nb _{0.1} Ru _{0.9} O ₂	1.69@1	100@0.3	Joule 7, 2023, 558.
RuCoO _x	1.56@0.2	10@0.1	J. Am. Chem. Soc. 2023, 145, 17995.
Y ₂ MnRuO ₇	1.75@1	24@0.2	Nat. Commun. 2023, 14, 2010.
W _{0.2} Er _{0.1} Ru _{0.7} O _{2-δ}	\	120@0.1	Nat. Commun. 2020, 11, 5368.
Ru/Co–N–C- 800 C	\	330@0.45	Adv. Mater. 2022, 34, 2110103.
Nd _{0.1} RuO _x /CC	1.595@0.05	50@0.01	Adv. Funct. Mater. 2023, 33, 2213304.
Ru _{0.6} Cr _{0.4} O ₂	\	12@0.1	Small Methods 2022, 6, 2200636.

Comments:

What is even more regrettable is that no detection or discussion of the key intermediate reactants of OER has been found in the entire manuscript, leaving important questions about the OER mechanism unclear.

Our response: We thank the reviewer for the valuable comment. We have gained further insights into the OER reaction mechanism. Detailed discussions have been provided below.

The discussion of OER real active species is the basis of OER reaction mechanism research. The lower current response in the CV curves corresponding to the oxidation-reduction of Ru in Ru/Ti₄O₇ compared to Ru and Ru/TiO implies the inhibited oxidation of Ru by Ti₄O₇ (**Supplementary Fig. 26**). **Supplementary Fig. 27** manifests that the Ru

and Ti loss of Ru/Ti₄O₇ gradually slows down during long-term OER. Ti gradually dissolves in the first 50 hours, which is attributed to the reaction between the unstable surface of the electrode. After 50 hours, the dissolution rate of Ti slows down due to the stabilization of the surface of the electrode. Furthermore, the characteristic diffraction peaks belonging to Ru and Ti₄O₇ can still be observed in the XRD pattern of Ru/Ti₄O₇ after OER without new characteristic peaks compared to the apparent dissolution of Ru/TiO (Supplementary Fig. 28). As shown in the electron paramagnetic resonance (EPR) spectra (Supplementary Fig. 29) and XPS spectra of O 1s before and after stability tests (Supplementary Fig. 30a), a slight decrease in oxygen vacancies in the Ti₄O₇ support is observed, indicating the structural stability of the Ti₄O₇ support under high current densities. In addition, the almost unchanged XPS signals of Ti and Ru with a slight shift towards high binding energy after OER further provide evidence for the good structural maintenance of Ru/Ti₄O₇ during the reaction (Supplementary Fig. 30b). The excellent stability of Ru/Ti₄O₇ is attributed to the inhibited oxidation process, which should be derived from the electron enrichment of Ru through the stabilized Ti–O–Ru units. The related details have been added in the revised manuscript, Page 11, Lines 203-205 and 208-213, highlighted in yellow.

Supplementary Fig. 26 CV curves of **a-d** Ru, Ru/TiO, Ru/Ti₄O₇, and Ru/TiO₂, respectively, without iR-corrected in argon-saturated 0.5 M H₂SO₄ at 50 mV s⁻¹ without iR compensation.

Supplementary Fig. 27 Dissolved Ru (left ordinate - y-axis) and Ti (right ordinate - y-axis) ion concentration in electrolyte for Ru/Ti₄O₇ determined via ICP-MS.

Supplementary Fig. 28 XRD patterns of a Ru/Ti₄O₇ after OER.

Supplementary Fig. 29 EPR spectra before and after the OER stability test at 200 mA cm^{-2} of $\text{Ru}/\text{Ti}_4\text{O}_7$.

Supplementary Fig. 30 XPS fine spectra of **a** O 1s and **b** Ti 2p with Ru 3p for $\text{Ru}/\text{Ti}_4\text{O}_7$ after OER.

A series of *in-situ* tests were performed to trace the origin of the activity and stability of $\text{Ru}/\text{Ti}_4\text{O}_7$. *In-situ* XAFS explored the structural evolution of $\text{Ru}/\text{Ti}_4\text{O}_7$ during OER. $\text{Ru}/\text{Ti}_4\text{O}_7$ underwent 20 CV cycles to obtain the stable surface before recording the *in-situ* XAFS spectra. In the most severe acidic environment, the intensity of the white line of the Ru *K*-edge increases with the applied potentials of OER (Fig. 5a), indicating the oxidation of Ru NPs. Furthermore, in the Fourier transform *R*-space (Fig. 5b), the gradually increased signal intensity of the Ru–O path verifies the formation of RuO_x . Notably, the peaks attributed to metallic Ru–Ru can always be observed even under the high potentials

of OER, which corresponds to the Ru metallic phase shown in the XRD pattern of Ru/Ti₄O₇ after OER. The above results indicate that the RuO_x species locally formed on the Ru/Ti₄O₇ can act as a barrier layer to slow down the further oxidation and dissolution of Ru NPs. The *in-situ* XAFS spectra show that the Ti *K*-edge of Ti₄O₇ almost completely overlaps under the OER potentials without prominent oxidation characteristics, which benefits from the stable structure of Ti₄O₇ (**Supplementary Fig. 31a**). The *R*-space of Ti further verifies that the Ti–O bond in Ti₄O₇ does not change significantly during OER (**Supplementary Fig. 31b**). The stability of Ru/Ti₄O₇ can be more verified in the explicit valence states of Ti, and Ru. The valence state corrected by the normalized white line intensity of the corresponding reference of Ti in Ru/Ti₄O₇ changes slightly between 3.52~3.61 with the application potentials growing (**Supplementary Fig. 32**). Meanwhile, the average valence state of Ru increases from 0.98 to 1.64. The significant change does not appear until the potential reaches 1.7 V vs. RHE (**Fig. 5c**). This phenomenon further verifies that the Ru NPs still maintain structural stability under high potentials of OER. The incomplete oxidation of Ru NPs is beneficial to maintaining the outstanding conductivity of the composite material during the electrocatalytic process. In the HRTEM image of Ru/Ti₄O₇ after OER cycling, the tight interfacial contact between Ru NPs and Ti₄O₇ can still be maintained (**Supplementary Fig. 33**). In addition, a locally amorphous structure can be observed in the outer layer of Ru NPs in Ru/Ti₄O₇ after OER, which should correspond to the RuO_x species (**Fig. 5d**).

Fig. 4 **a** Normalized *in-situ* Ru *K*-edge XANES of Ru/Ti₄O₇. **b** The corresponding *k*²-weighted Fourier transforms. **c** The valence states of Ru and Ti in Ru/Ti₄O₇ obtained via Ru *K*-edge under different potentials.

Supplementary Fig. 31 Normalized *in-situ* Ti K-edge XANES of Ru/Ti₄O₇. **b** The corresponding k^2 -weighted Fourier transforms.

Fig. R4 **a** TEM and **b-c** HRTEM images of Ru/Ti₄O₇ after OER cycling, respectively.

Moreover, the deeper reaction mechanism of OER was revealed by *in-situ* attenuated total reflection-surface enhanced infrared absorption spectra (ATR-SEIRAS) and a series of electrochemical tests.

Fig. 5e shows that, as the bias increases to 1.4 V vs. RHE, *in-situ* ATR-SEIRAS spectra of Ru exhibit absorption bands around the vibration frequency of 1038 cm⁻¹, corresponding to the *OOH/*OO intermediates. Simultaneously, a distinct absorption signal at the vibration frequency of 1189 cm⁻¹ gradually strengthens with the application of OER potential, corresponding to *O-O* in the LOM. *In-situ* ATR-SEIRAS spectra of Ru/Ti₄O₇ appear at similar positions as Ru, attributing to the absorption bands of *OOH/*OO (**Fig. 5f**). The broadening and shifting of absorption peaks may originate from the transition of the deprotonation process. Furthermore, almost no signal corresponding to *O-O* is observed around the vibration frequency of 1190 cm⁻¹.

Additionally, the normalized density difference of *in-situ* ATR-SEIRAS spectra corresponding to *OOH/*OO and *O-O* signals is presented in **Fig. 5g** to determine the proportion occupied by AEM and LOM in the acidic OER process [*J. Am. Chem. Soc.* 145, 2023, 23659] The higher adsorption density difference implies a higher proportion of AEM in the reaction process. In the high potential range, Ru undergoes surface reconstruction to form RuO_x species, enabling both AEM and LOM to drive OER. Meanwhile, *in-situ* ATR-SEIRAS spectra of Ru/Ti₄O₇ consistently exhibit a higher adsorption density difference than Ru during OER. This phenomenon indicates that Ru/Ti₄O₇ tends to drive OER through the AEM, which can realize more stable OER compared with the LOM with potential catalyst dissolution. In the pH-dependence tests, compared with Ru NPs, Ru/Ti₄O₇ showed less correlation with acidity, which verified that it was more inclined to drive OER with AEM (**Supplementary Fig. 34**). These results demonstrate that Ru/Ti₄O₇ tends to drive OER through the surface AEM, which can realize more stable OER compared with the LOM with potential catalyst dissolution.

Fig. 5 *In-situ* ATR-SEIRAS spectra of **e** Ru, and **f** Ru/Ti₄O₇ for OER in 0.5 M H₂SO₄ under different potentials vs. RHE. **g** Intensity difference of the infrared signals at 1038 and 1189 cm⁻¹.

Pulse voltammetry was employed to assess the deprotonation capability of catalysts, which can confirm the source of the enhanced activity of Ru/Ti₄O₇ in OER [*Nature* 587, 2020, 408; *Adv. Mater.* 35, 2023, 2208539]. Under different voltage pulses (**Supplementary Fig. 35**), Ru and Ru/Ti₄O₇ exhibit alternating cathodic and anodic current pulses (**Fig. 5h**). The oxidation charge storage capacity of different catalysts was further measured by integrating the anodic current response to voltage pulses. As shown

in **Fig. 5i**, Ru/Ti₄O₇ demonstrates a higher oxidation charge storage capacity compared to Ru NPs, implying that Ru/Ti₄O₇ undergoes a faster deprotonation process to form reaction intermediate *O [*Nature* 587, 2020, 408; *Adv. Mater.* 35, 2023, 2208539]. These findings suggest that the appropriate metal-support interactions between Ti₄O₇ and Ru can activate Ru sites by promoting the deprotonation process in OER. Furthermore, in the EIS bode plots, the pre-OER process of Ru/Ti₄O₇ results in an uneven distribution of surface charges (**Supplementary Fig. 36**), which is manifested by a reduction in frequency peaks within the range of 1.40-1.45 V vs. RHE and a shift towards higher frequencies compared to the broader transition phase peaks of Ru NPs (1.35-1.45 V vs. RHE). This phenomenon suggests that Ru/Ti₄O₇ exhibits a faster charge dissipation to accelerate deprotonation during OER to reaction kinetics [*J. Am. Chem. Soc.* 145, 2023, 23659]. In summary, the moderate metal-support interactions between Ru NPs and Ti₄O₇ through Ti–O–Ru units can reduce interface Schottky barriers to accelerate electron transfer and achieve electron enrichment at Ru sites to slow down the corrosion of Ru during OER. In addition, the electronic modulation of active Ru sites facilitates the deprotonation process of OER (**Fig. 5j**). The related details have been added in the revised manuscript, Pages 13-14, Lines 251-270; Page 14-15, Lines 273-285, highlighted in yellow.

Supplementary Fig. 34 LSV of a Ru and b Ru/Ti₄O₇ measured in different pH environments without iR compensation.

Supplementary Fig. 35 Pulse voltammetry protocol between 1.35 V cathodic and 1.42 V to 1.50 V vs. RHE anodic potentials without iR compensation.

Fig. 5 h Current responses to pulse voltammetry for Ru and Ru/Ti₄O₇. **i** Relationship between charge stored and potential for Ru and Ru/Ti₄O₇.

Supplementary Fig. 36 EIS Bode plots of a Ru and b Ru/Ti₄O₇ at the potentials of 1.20-1.60 V vs. RHE.

Comments:

The Fourier transform R-space assignment for Ru/Ti₄O₇ is weak and the evidence presented is weaker than claimed. On balance, it is recommended to reject the current work.

Our response: We thank the reviewer for the significant comment. The atoms used to form the Ti–O–Ru units are only the interface contact part between Ru and Ti₄O₇, which occupies a small ratio of Ru and Ti₄O₇. Therefore, the signal corresponding to the Ti–O–Ru units is weaker than the Ru–Ru signal in the *R* space of Ru *K*-edge in **Fig. 3e**. We have conducted a more detailed analysis of the Ru *K*-edge in *R* space and other characterizations to verify the formation of Ti–O–Ru interface units. In the *R* space of Ru *K*-edge of Ru/Ti₄O₇, the peak at 1.9 Å corresponds to the signal of Ru–O, which is different from the usual signal corresponding to Ru–O bonds located around 1.5 Å in Ru oxides [*Nat. Commun.* 2023, 14, 354; *J. Am. Chem. Soc.* 2021, 143, 17, 6482]. This signal should be attributed to the formation of Ti–O–Ru interface units between Ru NPs and Ti₄O₇ supports (**Fig. 3e**), which is consistent with the results reported in the literature [*Angew. Chem. Int. Ed.* 2022, e202116934; *Adv. Mater.* 2023, 2301369; *Angew. Chem. Int. Ed.* 2021, 60, 22276; *Nano Energy* 2021, 82, 105767]. Furthermore, we further confirmed the formation of Ti–O–Ru interface units through XPS. As shown in **Supplementary Fig. 8c**, Ru NPs exhibit only the characteristic peaks of metallic Ru. In the Ru 3d region of Ru/Ti₄O₇, the

signals attributed to Ru–O and Ru–Ru can be observed, corresponding to the interface Ti–O–Ru units [P. Natl. Acad. Sci. Usa. 2023, 120, e2312550120; Angew. Chem. Int. Ed. 2023, 62, e202305123]. In addition, in aberration-corrected high-resolution (AC-HRTEM) of Ru/Ti₄O₇, tight binding and appropriate matching at the nanoscale between the different phases can be observed, accompanied by a smooth transition at the interface (**Fig. R5**). The above information from *R*-space data combined with other experimental results can provide sufficient evidence for the formation of Ti–O–Ru units. The related details have been added in the revised manuscript, Page 8, Lines 140-142, highlighted in yellow.

Fig. 3e The k^2 -weighted Fourier transforms of EXAFS signals for Ru foil, Ru/TiO₂, Ru/Ti₄O₇, and Ru/TiO.

Supplementary Fig. 8 c XPS fine spectra of Ru 3*d* and C 1*s* of Ru, and Ru/Ti₄O₇,

respectively.

Fig. R5 a AC-HRTEM images of Ru/Ti₄O₇. b-c The schematic atom structure and the corresponding simulated HAADF-STEM images of Ti₄O₇ and Ru in Ru/Ti₄O₇, respectively. Blue, orange, and red spheres represent the Ti, Ru, and O atoms. d Schematic diagram of interface contact between Ti₄O₇ and Ru.

Reviewer #2:

Comments:

Peng et al. reported the construction of regulable supports via non-stoichiometric engineering to stabilize metal ruthenium for enhanced pH-universal water splitting. The Ru/Ti₄O₇ composite showed excellent performance for both the Hydrogen Evolution Reaction (HER) and Oxygen Evolution Reaction (OER) in pH-universal environments. Density Functional Theory (DFT) calculations were performed to provide insight into the origin of the good performance. Various types of calculations were carried out, including Pourbaix diagrams, band structures, density of states, and reaction profiles.

Our response: We appreciate the reviewer’s insightful and helpful comments on our manuscript. The reviewer's suggestions and criticisms help us substantially improve the quality of the manuscript. We have addressed the comments point-by-point as follows.

Comments:

However, it is still not clear how these calculations relate to the adsorption strength of different intermediates and, ultimately, the performance.

Our response: We thank the reviewer for the great effort in improving our work. We apologize for the confusion caused by the unclear presentation of data to the reviewers. In **Supplementary Table 5 and 7**, we have provided detailed adsorption energy values of the reaction intermediates in the OER and HER. We also calculated the difference in adsorption free energy to clarify the relationship between reaction performance and the adsorption energy of the reaction intermediates (**Supplementary Table 6**). The related discussions have been added in the Supplementary Information, Page 59, Supplementary Table 5-7, highlighted in yellow.

Supplementary Table 5. Adsorption free energy of OER intermediates at U=0.

Catalysts	ΔG^{*OH} (eV)	ΔG^{*O} (eV)	ΔG^{*OOH} (eV)
Ru/Ti ₄ O ₇	0.703	1.950	3.380
Ru/TiO ₂	0.435	1.977	3.315
Ru/TiO	-0.482	0.808	2.624

Supplementary Table 6. Adsorption free energy changes of OER intermediates at U=0.

Catalysts	ΔG_1 (eV)	ΔG_2 (eV)	ΔG_3 (eV)	ΔG_4 (eV)	ΔG (eV)
Ru/Ti ₄ O ₇	0.703	1.247	1.431	1.540	1.540
Ru/TiO ₂	0.435	1.541	1.338	1.605	1.605
Ru/TiO	-0.482	1.290	1.816	2.296	2.296

Supplementary Table 7. Adsorption free energy of HER intermediates at U=0.

Catalysts	ΔG^*_{H} (eV)
Ru	-0.745
Ru/Ti ₄ O ₇	0.123
Ru/TiO ₂	-0.171
Ru/TiO	-0.446

Comments:

More detailed analysis and computational details are required for a clear understanding.

Our response: We thank the reviewer for the valuable suggestion to improve our manuscript. According to the suggestion, we have supplemented the manuscript with additional detailed analysis and computational details to enhance the clarity and understanding of the corresponding content. The related details have been added in the revised manuscript, Page 17, Lines 326-336 and 339-346; Page 24, Lines 500-503, highlighted in yellow.

Comments:

Importantly, I have stronger concerns about the chemical structure models employed to study the reaction mechanisms. Since it is an interface model, which surfaces of TiO₂, Ti₄O₇, and TiO were used to contact with what surface of Ru? This information is crucial to know if the interfaces have a proper lattice match and if the right surfaces were used with respect to the experimental ones.

Our response: We thank the reviewer for the constructive comments. The crystal facets are distinct for each titanium oxide support to form the heterogeneous interfaces with Ru NPs due to the different crystal structures of TiO₂, Ti₄O₇, and TiO. In addition, we attempted to create heterogeneous structures with the same Ru crystal facets. However, it is challenging to ensure adequate lattice matching between Ru and different titanium oxides. Therefore, we determined the following interface information to ensure the lattice mismatch below 5% and expose conventional crystal facets of Ru NPs. As shown in **Fig. R6**, the (0 0 1) facet of TiO contacts the (0 1 0) facet of Ru. The (1 -1 0) facet of TiO₂

contacts the (2 1 0) facet of Ru. The (1 1 -1) facet of Ti₄O₇ contacts the (0 0 1) facet of Ru. The related details have been added in the revised manuscript, Page 24, Lines 500-503, highlighted in yellow.

Fig. R6 Relaxation structure of **a** Ru, **b** Ru/Ti₄O₇, **c** Ru/TiO₂, and **d** Ru/TiO used to calculate OER reaction steps.

Comments:

Additionally, how did the authors determine the most stable models? The author used a sandwich interface model (Fig 30c in SI) and claimed charge accumulation on Ru atoms, connecting it with the performance. However, this contact is different from the interface model (Fig S 31-33 in SI) for HER and OER analysis. These two different models have different exposed sites with different chemical environments. Consequently, it is not suitable to be used together to explain the good performance. The surface states of Ru in Fig S 31-33 in SI are different, creating a new factor that affects the adsorption strength of *O, *OH, and *OOH. This negatively affects the arguments on the lower overpotential due to the Ru/Ti₄O₇ interface.

Our response: We thank the reviewer for the thoughtful and detailed suggestion. The material surface undergoes reconstruction into different active phases during HER and

OER. To simulate reaction states more accurately, we retained the clean surface of Ru-TiO_x to simulate the reduced state in HER. For OER, experimental results demonstrate the active phase after OER, serving as direct evidence for constructing models for the OER reaction. Therefore, during OER, we performed local oxidation on the surface of the heterojunction model to simulate the oxidized state of Ru, which is a common practice in OER studies [*Angew. Chem. Int. Ed.* 2022, 61, e202207217; *Nat. Commun.* 2021, 12, 3540].

As the reviewer commented, the Ru active sites on the surface exhibited distinct chemical environments due to the applied biases of OER and HER. However, electron transfer still applies to the inside of the material, even under different surface reduction/oxidation states. Therefore, the differential metal-support interactions between Ru and various titanium oxides still influence the surface oxidation structure after the OER. To validate this viewpoint further, we have provided additional calculations of the DOS and *d*-band centers of the structural model with surface oxidation and discussed the correlation with OER activity (**Fig. R7**). The related discussions have also been added in the revised manuscript, Page 17, Lines 326-336, highlighted in yellow.

The local oxidation combined with the electron redistribution between Ru and different titanium oxide supports endow the Ru sites with varying intrinsic activities through the *d*-band center modulation. The downward shift of the *d*-band center reduces the energy of the antibonding orbitals formed by the adsorption reaction intermediates and the *d*-orbitals of Ru, which implies the weak adsorption of reaction intermediates [*Adv. Energy Mater.* 2023, 13, 2300177; *Adv. Funct. Mater.* 2022, 32]. Therefore, in the surface oxidized Ru/TiO, Ru/TiO₂ and Ru/Ti₄O₇, the *d*-band centers of the Ru sites are -1.80 eV, -1.81 eV and -2.07 eV, respectively (Supplementary Fig. 44 and Supplementary Table 5), which endow the corresponding *OH adsorption free energies with -0.482 eV, 0.435 eV, and 0.703 eV at U=0 V, respectively (Fig. 6d). Subsequently, the deprotonation process is also optimized. Ru/Ti₄O₇ exhibits the smallest deprotonation free energy and the lowest theoretical potential for OER ($\Delta G = 1.540$ eV, Supplementary Table 6). The enhanced deprotonation process is consistent with experimental results.

Fig. R7 a Projected density of states (PDOS) of Ru 3d in the relaxation structures of Ru, Ru/Ti₄O₇, Ru/TiO₂, and Ru/TiO used to calculate OER reaction steps. **b** Free energy profiles of different OER intermediates at 0 V and 1.23 V for Ru site in Ru, Ru/TiO₂, Ru/Ti₄O₇, and Ru/TiO.

Comments:

The author claimed that electron migration between Ru and titanium oxide forms electron-rich Ru sites and weakens the adsorption of the HER intermediates. This is a very general comment for the Ru/TiO₂, Ru/Ti₄O₇, and Ru/TiO, leaving the question of why Ru/Ti₄O₇ had better performance unanswered.

Our response: We thank the reviewer for the valuable comment. We have provided a more detailed explanation of the metal-support interactions between Ru and different titanium oxides for the activation of Ru activity in HER. The electron migration between Ru and the supports directly acts on the *d*-orbital electrons of Ru, which the change of the *d*-band center can describe. Researchers widely adopt the *d*-band center as a mature theoretical approach to predict or explain the activity of catalysts [*Nature* 1995, 376, 238–240; *J. Mol. Catal. A: Chem.* 1997, 115, 421; *Angew. Chem. Int. Ed.* 2020, 59, 18036]. Therefore, we focused on the changes in the *d*-band center of Ru atoms at the interface of Ru/TiO, Ru/TiO₂, and Ru/Ti₄O₇ and used the *d*-band center theory to interpret the differences in HER activity. As shown in **Fig. R8a**, the *d*-band center of Ru at the interface exhibits distinct variations with the interface electron migration induced by the contact between Ru and different titanium oxides. The *d*-band center of Ru is -1.72 eV. After contact with TiO, TiO₂, and Ti₄O₇, the corresponding *d*-band centers of Ru shift to -1.77 eV, -1.89 eV, and -1.98 eV, respectively. The downward shift of the *d*-band center

reduces the energy of the antibonding orbitals formed by the adsorption reaction intermediates, and the *d*-orbitals of Ru, which implies the weak adsorption of reaction intermediates [*Nature* 1995, 376, 238–240; *J. Mol. Catal. A: Chem.* 1997, 115, 421]. Therefore, the metal-support interactions between Ru and different titanium oxides can weaken the strong adsorption of HER intermediates on Ru sites to enhance the intrinsic activity, which is consistent with previous reports [*Nat. Commun.* 2022, 13, 5382; *Angew. Chem. Int. Ed.* 2022, 61, e202212196; *Angew. Chem. Int. Ed.* 2022, 61, e202209486]. Owing to the various *d*-band center induced by the metal-support interaction, the Ru sites in Ru, Ru/TiO, Ru/TiO₂, and Ru/Ti₄O₇ have *H adsorption free energies of –0.745 eV, –0.446 eV, –0.171 eV, and 0.123 eV, respectively (**Fig. R8b, Supplementary Table 7**). Therefore, Ru/Ti₄O₇ has the lowest theoretical overpotential of 0.123 eV. The related discussions have also been added in the revised manuscript, Page 17, Lines 339-346, highlighted in yellow.

Specifically, the *d*-band center of Ru at the interface exhibits distinct variations with the interface electron migration induced by the contact between Ru and different titanium oxides. The *d*-band center of Ru is –1.72 eV. After contact with TiO, TiO₂, and Ti₄O₇, the corresponding *d*-band centers of Ru shift to –1.77 eV, –1.89 eV, and –1.98 eV, respectively (Supplementary Fig. 46). Owing to the downward shift compared to the original Ru NPs induced by the metal-support interaction, the Ru sites in Ru, Ru/TiO, Ru/TiO₂, and Ru/Ti₄O₇ have *H adsorption free energies of –0.745 eV, –0.446 eV, –0.171 eV, and 0.123 eV, respectively (Fig. 6e, Supplementary Table 7) [*Nature* 1995, 376, 238–240; *J. Mol. Catal. A: Chem.* 1997, 115, 421]. Therefore, Ru/Ti₄O₇ has the lowest theoretical reaction potential of 0.123 eV for HER.

Fig. R8 a PDOS of Ru 3d in the relaxation structures of Ru, Ru/Ti₄O₇, Ru/TiO₂, and Ru/TiO used to calculate HER reaction steps. **b** Free energy profiles of HER intermediates for Ru site in Ru, Ru/TiO₂, Ru/Ti₄O₇, and Ru/TiO.

Reviewer #3:

Comments:

The authors have developed a Ti₄O₇-loading Ru nanoparticles catalyst for pH-universal water splitting. Ru/Ti₄O₇ exhibits ultralow overpotentials of 8 mV and 150 mV for HER and OER, respectively, maintaining stable performance over 500 hours at 10 mA cm⁻² in acidic media.

I thank the authors for demonstrating the catalyst's outstanding activities. However, the stability concerns raised in the manuscript need to be addressed, major revisions are recommended before being considered for publication in Nature Communications.

Our response: We thank the reviewer for the positive evaluation of our manuscript. We have revised the manuscript according to your valuable comments and suggestions.

1. In the manuscript (line 93), the authors state, "The suitable oxygen vacancies in Ti₄O₇ promote the Ru³⁺ adsorption on Ti₄O₇, which endows the Ru NPs with a more uniform distribution on Ti₄O₇ compared to TiO₂ with low vacancies." I am puzzled by whether Ru can attach to oxygen vacancies when there is no oxygen present in these vacancies. How

does Ru bond to Ti in the absence of oxygen?

Our response: We thank the reviewer for the great comment. The adsorption of cations based on oxygen vacancies is achieved through coordination with neighboring oxygen around the oxygen vacancies, as shown in **Fig. R9** [*Adv. Funct. Mater.* 2023, 33, 2300673]. In comparison to the clean surface, the coordination of neighboring oxygen with metal cations near oxygen vacancies requires lower energy, as confirmed and extensively studied in previous reports [*Nat. Commun.* 2022, 13, 2473; *Adv. Funct. Mater.* 2023, 33, 2300673]. In our work, firstly, Ru^{3+} was adsorbed on the surface of Ti_4O_7 by the adjacent oxygen near the oxygen vacancies. Subsequently, the adsorbed Ru was the nucleation center, and Ru gradually grew into nanoparticles (NPs) with the increased Ru^{3+} concentration and annealing in high-temperature environments. Finally, the Ru NPs contacted the broader surface of the Ti_4O_7 support to form interface Ti–O–Ru units (**Fig. R10**). Therefore, the vicinity of the oxygen vacancy only serves as a nucleation center during the formation of Ru NPs. The formation of the Ti–O–Ru units interface is based on the extensive contact between Ru NPs and the Ti_4O_7 , which is not only limited to local oxygen vacancies. In summary, the existence of interface units is reasonable.

Fig. R9 Schematic diagram of the adsorption of Ru cations by ZrO_{2-x} with O vacancies [*Adv. Funct. Mater.* 2023, 33, 2300673].

Fig. R10 Relaxation structure of Ru/Ti₄O₇.

Furthermore, the formation of Ti–O–Ru interface units is supported by sufficient data. We have conducted a more detailed analysis of the Ru *K*-edge in *R* space and other characterizations to verify the formation of Ti–O–Ru interface units. In the *R* space of Ru *K*-edge of Ru/Ti₄O₇, the peak at 1.9 Å corresponds to the signal of Ru–O, which is different from the usual signal corresponding to Ru–O bonds located around 1.5 Å in Ru oxides [*Nat. Commun.* 2023, 14, 354; *J. Am. Chem. Soc.* 2021, 143, 17, 6482]. This signal should be attributed to the formation of Ti–O–Ru interface units between Ru NPs and Ti₄O₇ supports (**Fig. 3e**), which is consistent with the results reported in the literature [*Angew. Chem. Int. Ed.* 2022, e202116934; *Adv. Mater.* 2023, 2301369; *Angew. Chem. Int. Ed.* 2021, 60, 22276; *Nano Energy* 2021, 82, 105767]. Furthermore, we further confirmed the formation of Ti–O–Ru interface units through XPS. As shown in **Supplementary Fig. 8c**, Ru NPs exhibit only the characteristic peaks of metallic Ru. In the Ru 3d region of Ru/Ti₄O₇, the signals attributed to Ru–O and Ru–Ru can be observed, corresponding to the interface Ti–O–Ru units [*P. Natl. Acad. Sci. USA.* 2023, 120, e2312550120; *Angew. Chem. Int. Ed.* 2023, 62, e202305123]. In addition, in aberration-corrected high-resolution (AC-HRTEM) of Ru/Ti₄O₇, tight binding and appropriate matching at the nanoscale between the different phases can be observed, accompanied by a smooth transition at the interface (**Fig. R5**). The above information from *R*-space data combined with other experimental results can provide sufficient evidence for the formation of Ti–O–Ru units. The related details have been added in the revised manuscript, Page 8, Lines 140-142, highlighted in yellow.

Fig. 3e The k^2 -weighted Fourier transforms of EXAFS signals for Ru foil, Ru/TiO₂, Ru/Ti₄O₇, and Ru/TiO.

Supplementary Fig. 8 c XPS fine spectra of Ru 3d and C 1s of Ru, and Ru/Ti₄O₇, respectively.

Fig. R5 a AC-HRTEM images of Ru/Ti₄O₇. b-c The schematic atom structure and the corresponding simulated HAADF-STEM images of Ti₄O₇ and Ru in Ru/Ti₄O₇, respectively. Blue, orange, and red spheres represent the Ti, Ru, and O atoms. d Schematic diagram of interface contact between Ti₄O₇ and Ru.

2. Ti₄O₇ possesses oxygen vacancies, can these vacancies remain stable under high-current OER conditions? In extensive oxygen-deficient materials, defects can be filled with reactive oxygen during the OER process. Can the structure of Ti₄O₇ withstand this, especially considering the observed continuous leaching of basal Ti in Fig. 4f? Authors are encouraged to provide evidence.

Our response: We thank the reviewer for the good suggestion. To validate the structural characteristics of the Ti₄O₇ support under high current density conditions, we conducted OER stability tests at high current densities [*Nat. Commun.*, 2023, 14, 843; *Nat. Commun.*, 2020, 11, 5368]. **Supplementary Fig. 24** presents the chronopotentiometry data for Ru/Ti₄O₇ at a high current density of 200 mA cm⁻². The potential of Ru/Ti₄O₇ remained steady for 300 h. Furthermore, as shown in the electron paramagnetic resonance (EPR)

spectra (**Supplementary Fig. 29**) and XPS spectra of O 1s (**Supplementary Fig. 30**) before and after the stability test, a slight decrease in oxygen vacancies in the Ti₄O₇ support is observed, indicating the structural stability of the Ti₄O₇ support under high current densities. This result can be comparable to that of recently reported Ru-based catalysts [*Nat. Commun.*, 2023, 14, 354; *Nat. Commun.*, 2022, 13, 3784; *Nat. Commun.*, 2023, 14, 1412]. The related details have been added in the revised manuscript, Page 11, Lines 208-213, highlighted in yellow.

Supplementary Fig. 24 Chronopotentiometry curves of Ru/Ti₄O₇ at 200 mA cm⁻² in 0.5 M H₂SO₄.

Supplementary Fig. 29 EPR spectra before and after the OER stability test at 200 mA cm⁻² of Ru/Ti₄O₇.

Supplementary Fig. 30 XPS fine spectra of **a** O 1s and **b** Ti 2p with Ru 3p for Ru/Ti₄O₇ after OER.

Additionally, we characterized the Ti dissolution during OER through ICP-MS. As shown in **Supplementary Fig. 27**, Ti gradually dissolves in the first 50 hours, which is attributed to the reaction between the unstable surface of the electrode. After 50 hours, the dissolution rate of Ti slows down due to the stabilization of the surface of the electrode. In summary, the Ti₄O₇ support maintains its structure under high current density conditions, which induces the appropriate metal-support interactions to simultaneously enhance the catalytic activity and stability of Ru/Ti₄O₇. The related details have been added in the revised manuscript, Page 11, Lines 203-205, highlighted in yellow.

Supplementary Fig. 27 Dissolved Ru (left ordinate - y-axis) and Ti (right ordinate - y-axis) ion concentration in electrolyte for Ru/Ti₄O₇ determined via ICP-MS.

3. The authors emphasize "Interface crystal structure matching between Ti_4O_7 and Ru forms Ti–O–Ru units to stabilize Ru species during OER." Why was stability not measured under high-current acidic conditions, and only at 10 mA/cm^{-2} ? The stability decay rate at 10 mA/cm^{-2} (maybe $142 \mu\text{V/h}$), calculated from Fig. 3h and Fig. 6b, can definitely not demonstrate its stability.

Our response: We thank the reviewer for this valuable question. To validate the OER stability of Ru/ Ti_4O_7 under high current densities, we conducted chronoamperometric stability tests at a high current density of 200 mA cm^{-2} . As shown in **Supplementary Fig. 24**, Ru/ Ti_4O_7 shows a slight performance degradation to achieve 200 mA cm^{-2} for 300 h in 0.5 M H_2SO_4 . Inspired by the excellent stability of Ru/ Ti_4O_7 in the three-electrode system, the performance of the PEM electrolyzer assembled with Ru/ Ti_4O_7 was further evaluated. The PEM electrolyzer with Ru/ Ti_4O_7 as both anode and cathode well maintained its voltage around 1.74 V during 300 h operation at 500 mA cm^{-2} (**Fig. R2**). Ru/ Ti_4O_7 also exhibits excellent PEM performance compared with recently reported catalysts, revealing its potential for practical applications (**Supplementary Table 9**). In summary, Ru/ Ti_4O_7 has excellent stability under high current densities. The related details have been added in the revised manuscript, Page 10, Lines 198-200; Page 19, Line 369-370, highlighted in yellow.

Supplementary Fig. 24 Chronopotentiometry curves of Ru/ Ti_4O_7 at 200 mA cm^{-2} in 0.5 M H_2SO_4 .

Fig. R2 a Conceptual model of the PEM water electrolyzer flow cell using pure water as feedstock. **b** Stability tests of Ru/Ti₄O₇ || Ru/Ti₄O₇ for PEM water electrolyzer at 500 mA cm⁻².

Supplementary Table 9. Comparison of the PEM performance of this work with that of the Ru based catalysts recently reported.

Catalysts	E (V)@J (A cm ⁻²)	Stability (h) @J (A cm ⁻²)	Reference
Ru/Ti₄O₇	1.69@0.5	300@0.5	This work
Ni-RuO ₂	1.78@0.5	1000@0.2	Nat. Mater. 2023, 22, 100.
SS Pt-RuO ₂ HNSs	\	100@0.1	Sci. Adv. 2022, 8, eabl9271.
Nb _{0.1} Ru _{0.9} O ₂	1.69@1	100@0.3	Joule 7, 2023, 558.
RuCoO _x	1.56@0.2	10@0.1	J. Am. Chem. Soc. 2023, 145, 17995.
Y ₂ MnRuO ₇	1.75@1	24@0.2	Nat. Commun. 2023, 14, 2010.
W _{0.2} Er _{0.1} Ru _{0.7} O _{2-δ}	\	120@0.1	Nat. Commun. 2020, 11, 5368.
Ru/Co-N-C-800 C	\	330@0.45	Adv. Mater. 2022, 34, 2110103.
Nd _{0.1} RuO _x /CC	1.595@0.05	50@0.01	Adv. Funct. Mater. 2023, 33, 2213304.
Ru _{0.6} Cr _{0.4} O ₂	\	12@0.1	Small Methods 2022, 6, 2200636.

4. It is recommended that the authors supplement the manuscript with CV after the OER stability test, along with ICP of Ru in the catalyst and XPS data. In Fig. 4f, for the Ru solution ICP, a platform was not observed but an increase, and it is suggested that the authors extend the testing time to 100 hours.

Our response: We thank the reviewer for the significant comment. Detecting catalyst mass losses during OER can provide quantitative information that distinguishes different degradation mechanisms. We characterized the long-term stability-induced variations in Ti dissolution through ICP-MS. As shown in **Supplementary Fig. 27**, Ti gradually dissolves in the first 50 hours, which is attributed to the reaction between the unstable support and the electrolyte. After 50 hours, the dissolution rate of Ti slows down due to the stabilization of the support surface, where the metal-support interaction inhibits the dissolution of Ti and Ru. In summary, the dissolution rates of Ru and Ti in Ru/Ti₄O₇ decreased over time, demonstrating a stable structure during OER. The results of the CV experiments further highlight that Ru/Ti₄O₇ exhibits a negligible increase of overpotential after 3000 cycles (**Supplementary Fig. 25**). The chronopotentiometry and CV measurements demonstrate that Ti₄O₇ support can stabilize the Ru sites in harsh environments during OER. Furthermore, we have added the XPS spectra of Ru/Ti₄O₇ after OER. The great maintenance of Ti and Ru signals before and after OER further suggested the structural stability of Ru/Ti₄O₇ (**Supplementary Fig. 30**). The related details have been added in the revised manuscript, Page 10, Lines 188-190 and 198-200; Pages 11, Lines 203-205, highlighted in yellow.

Supplementary Fig. 27 Dissolved Ru (left ordinate - y-axis) and Ti (right ordinate - y-axis) ion concentration in electrolyte for Ru/Ti₄O₇ determined via ICP-MS.

Supplementary Fig. 25 OER CV curves of Ru/Ti₄O₇ after 3000th cycles.

Supplementary Fig. 30 XPS fine spectra of **a** O 1s and **b** Ti 2p with Ru 3p for Ru/Ti₄O₇ after OER.

5. I am curious about the reason behind the bipolar stability being lower than MEA stability, especially when the stability at 10 mA/cm⁻² is evidently decaying in the article. Why does the stability improve in MEA?

Our response: We thank the reviewer for this valuable comment. We will elaborate on the following aspects. (1) Different electrolytes. In the dual-electrode systems, we selected

0.5 M H₂SO₄ (pH = 0.3) as the electrolyte to assess the stability of Ru/Ti₄O₇. However, due to the strong adsorption of SO₄²⁻ on the catalyst surface, the valence state of the active site increases, which accelerates the corrosion of the catalyst and causes activity decay [*ChemPhysChem* 2019, 20, 2956]. In contrast, the electrolyte used in the PEM electrolyzer is pure water, where the acidic environment is caused by protons generated at the anode during the electrolysis of water, and the pH value is maintained around 2-3 during stable PEM operation. Therefore, the relatively mild electrolyte environment in the PEM electrolyzer favors the long-term stable operation of Ru/Ti₄O₇. (2) Different membranes. PEM electrolyzers use proton exchange membranes to separate the anode and cathode sides. The membrane selectively transports protons and prevents the mutual diffusion of hydrogen and oxygen, which reduces the mixing of hydrogen and oxygen within the dual-electrode electrolyzer to prevent adverse reactions and improve system stability [*Science* 2023, 380, 609; *Adv. Mater.* 2021, 2006328; *Aggregate.* 2021, 2, e106]. (3) Different operating environments. PEM electrolyzers use a flowing electrolyte and operate at 80 °C, unlike the dual-electrode system under ambient conditions [*Joule* 2023, 7, 558; *Energy Environ. Sci.*, 2020, 13, 5143]. The optimization of test conditions accelerates the reaction kinetics of Ru/Ti₄O₇, contributing to performance improvement. In summary, the optimization of electrolyte, membrane, and operating environment in the PEM electrolyzer enhances the stability of Ru/Ti₄O₇ compared to the dual-electrode system.

6. In Fig. 5b, it is noticeable that Ru is more prone to dissolution in a pH=14 environment. Why does stability in an AEM (alkaline environment) (500 mA/cm⁻²) surpass that in a PEM (200 mA/cm⁻²)?

Our response: We thank the reviewer for this valuable comment. The change rate of the cell voltage should be the main indicator to evaluate the stability of the devices of water electrolysis by chronopotentiometry rather than the operating time [*Nat. Commun.* 2023, 14, 843 (2023)]. Therefore, we have described the change rate of cell voltage for the AEM and PEM in this work. We have further added the stability data of the PEM at a current density of 500 mA cm⁻². Specifically, at a current density of 200 mA cm⁻², the PEM electrolyzer shows the rate of rise of 0.025 mV h⁻¹ (**Fig. 7f**). In addition, at a current

density of 500 mA cm^{-2} , the PEM electrolyzer shows the rate of rise of 0.03 mV h^{-1} (**Fig. R2**), while the AEM electrolyzer shows the rate of rise of about 1.23 mV h^{-1} (**Supplementary Fig. 51**). Experimental results indicate that the PEM electrolyzer has slower performance decay than that of AEM.

Supplementary Fig. 51 a Steady polarization curves of Ru/Ti₄O₇ || Ru/Ti₄O₇ and RuO₂ || Pt/C for anion exchange membrane (AEM) water electrolyzer using 1 M KOH as feedstock. **b** Chronopotentiometry tests of Ru/Ti₄O₇ || Ru/Ti₄O₇ for AEM water electrolyzer at 500 mA cm^{-2} .

Fig. 7. e Steady polarization curves of Ru/Ti₄O₇ || Ru/Ti₄O₇ and RuO₂ || Pt/C for PEM water electrolyzer. **f** Stability tests of Ru/Ti₄O₇ || Ru/Ti₄O₇ for PEM water electrolyzer at 200 mA cm^{-2} .

Fig. R2 a Conceptual model of the PEM water electrolyzer flow cell using pure water as feedstock. **b** Stability tests of Ru/Ti₄O₇ || Ru/Ti₄O₇ for PEM water electrolyzer at 500 mA cm⁻².

In addition, Fig. 6b only reflects the ideal state of Ru, which can be used as a reference for phase analysis. The characteristics of the catalysts are not the only factor for the performance of the device. The AEM and PEM electrolyzers have different compositional structures in this work. Specifically, PEM (Nafion 117) has stronger mechanical strength and corrosion resistance at high temperatures than AEM (PiperION-A60). Due to the weak mechanical strength of AEM, the catalysts were coated on the gas diffusion layer to avoid swelling and perforation of the AEM during the membrane electrode preparation. For PEM electrolyzers, the catalysts can be directly formed into a membrane electrode through transfer printing and hot pressing. These factors together contribute to the performance differences between AEM and PEM in this work.

7. In Line 142, "reduced valence" possibly an error, should it be "oxide valence"?

Our response: We thank the reviewer for the reminder. We apologize for the confusion caused by the unclear presentation. We have changed the corresponding description. The related discussions have also been added in the revised manuscript, Page 8, Line 158, highlighted in yellow.

As shown in Fig. 2g, the shift of Ti pre-K-edge reflects the changed **oxide valence** states of Ti in different stoichiometric titanium oxide supports.

8. In Line 173, SI Table 1 is mentioned as mass activity assemble, but it is actually ICP-MS data.

Our response: We thank the reviewer for the good comment. We apologize for the confusion caused by the unclear presentation. Herein, we illustrate that mass activity is obtained based on ICP-MS data. We have modified the manuscript and provided additional clarifications in the Supplementary Information, Page 22.

The LSV polarization curves normalized by C_{Ru} and the mass of Ru indicate that the active Ru sites in Ru/Ti₄O₇ have the highest specific activities (Supplementary Fig. 20) and mass activity (Supplementary Fig. 21).

Supplementary Fig. 21 Mass activity of different catalysts for **a** OER and **b** HER, respectively.

The Ru content was confirmed via inductively coupled plasma-mass spectrometry (ICP-MS), as shown in Supplementary Table 1, to calculate the mass activity of different catalysts.

At last, we wish to thank the Editor and the Reviewers again for the very constructive comments and suggestions to improve the quality of our manuscript. Thank you very much!

REVIEWERS' COMMENTS

Reviewer #1 (Remarks to the Author):

The manuscript has been carefully revised. The revised manuscript completely resolves my confusion and seems ready for publication.

Reviewer #2 (Remarks to the Author):

My comments have satisfactorily addressed most of my comments. It is recommended to be accepted for publication.

Reviewer #3 (Remarks to the Author):

The development of proton exchange membrane (PEM) water electrolyzers is crucial for improving hydrogen production efficiency and represents a key pathway for hydrogen generation. In this work, Peng et al. have prepared Ru/Ti₄O₇, demonstrating exceptional OER/HER performance in pH-universal electrolytes. The PEM assembled by the authors showed impressive performance. I would like to recommend this manuscript for publication in Nature Communications. To enhance the manuscript, the following points should be addressed:

1. Figure 7 illustrates that Ru/Ti₄O₇ exhibits greater stability at 200 mA cm⁻² compared to 10 mA cm⁻². It would be beneficial to clarify whether this difference is influenced by the operating temperature of the PEM.
2. Previous studies, such as Liu et al. (Joule 7, 558-573), have indicated that doping specific metal elements into RuO₂ can prevent the reaction from the LOM path. The authors employed a loading approach for synthesizing this material, resulting in limited Ti-O-Ru sites. It would be valuable to discuss the advantages of this loaded catalyst from a material design perspective.

Manuscript ID: NCOMMS-23-53716A-Z
Point-by-Point Response to Review Comments

We sincerely appreciate the pertinent suggestions of editors and reviewers. The quality of the manuscript has been significantly improved following reviewers' comments. All modifications have been highlighted with yellow shading in the revised manuscript.

Reviewers' comments:

Reviewer #1:

Comments:

The manuscript has been carefully revised. The revised manuscript completely resolves my confusion and seems ready for publication.

Our response: We thank the reviewer for acknowledging the acceptance of our work.

Reviewer #2:

Comments:

My comments have satisfactorily addressed most of my comments. It is recommended to be accepted for publication.

Our response: We thank the reviewer for acknowledging the acceptance of our work.

Reviewer #3:

Comments:

The development of proton exchange membrane (PEM) water electrolyzers is crucial for improving hydrogen production efficiency and represents a key pathway for hydrogen generation. In this work, Peng et al. have prepared Ru/Ti₄O₇, demonstrating exceptional OER/HER performance in pH-universal electrolytes. The PEM assembled by the authors

showed impressive performance. I would like to recommend this manuscript for publication in Nature Communications. To enhance the manuscript, the following points should be addressed:

1. Figure 7 illustrates that Ru/Ti₄O₇ exhibits greater stability at 200 mA cm⁻² compared to 10 mA cm⁻². It would be beneficial to clarify whether this difference is influenced by the operating temperature of the PEM.

Our response: We thank the reviewer for the significant suggestion. The structure of the PEM electrolyzer is different from the two-electrode electrolysis cell. In the PEM electrolyzer, the catalyst layer closely contacts the PEM to form a zero-gap catalytic electrode, which can reduce the solution resistance to lower the cell voltage. Additionally, previous studies indicate that, under the same operating current density, elevated temperatures can expedite proton transport in the PEM, effectively reducing the cell voltage [Adv. Mater. 2024, 2305711; ACS Catal. 2023, 13, 7568–7577]. This phenomenon diminishes the corrosion of the PEM anode caused by the high cell voltage. Therefore, the unique structural advantages of the PEM electrolyzer and the characteristics of high-temperature operation contribute beneficially to the stability of the PEM water electrolysis compared to the two-electrode system tested at room temperature. The related discussions have also been added in the revised manuscript, Page 19, Lines 373-375, highlighted in yellow.

2. Previous studies, such as Liu et al. (Joule 7, 558-573), have indicated that doping specific metal elements into RuO₂ can prevent the reaction from the LOM path. The authors employed a loading approach for synthesizing this material, resulting in limited Ti-O-Ru sites. It would be valuable to discuss the advantages of this loaded catalyst from a material design perspective.

Our response: We thank the reviewer for the valuable comment. In response to the reviewer's suggestions, we discuss the advantages of loaded catalysts from a material design perspective. Doped catalysts require suitable heteroatoms and precise doping amounts to ensure the incorporation of heteroatoms into the material lattice rather than forming heterogeneous structures. Therefore, the design of doped catalysts is based on

meticulous structural control and a complex preparation process. Additionally, heteroatom doping can alter the band structure of the materials, which may not always favor the conductivity [J. Electroanal. Chem. 2020, 878, 114593]. In comparison, loaded catalysts have the following advantages: (1) The loaded catalysts are easy to synthesize, enabling scalable production for industrial applications [Small Methods 2022, 6, 2200855]. (2) The mass content of active species in the loaded catalysts is typically below 50%. The active phases can be evenly dispersed on the supports, which facilitates the sufficient exposure of active sites to enhance the atomic utilization efficiency [Nat. Commun. 2023, 14, 5119; Angew. Chem. Int. Ed. 2023, 62, e202305186]. (3) Loaded catalysts can effectively reduce the usage of precious metals as the active species [Angew. Chem. Int. Ed. 2022, 61, e202212341]. (4) The highly conductive supports are always advantageous for conductivity improvement of the loaded catalysts. (5) Sufficient contact between the support and active phase can construct suitable metal-support interactions and stable interface units to regulate the coordination configuration and electronic structure of interface sites, which optimizes the catalytic performance and stability of the active centers [Science, 2023, 380, 644-651; Nat. Commun. 2024, 15, 559]. The related discussions have also been added in the revised manuscript, Page 3, Lines 45-51, highlighted in yellow.

At last, we wish to thank the Editor and the Reviewers again for the very constructive comments and suggestions to improve the quality of our manuscript. Thank you very much!